# τ-Knowledge: Evaluating Conversational Agents over Unstructured Knowledge

**Quan Shi** [* 1]  **Alexandra Zytek** [* 1]  **Pedram Razavi** [1]  **Karthik Narasimhan** [2]  **Victor Barres** [1]

 Code      Leaderboard      Visualizer

## Abstract

Conversational agents are increasingly deployed in knowledge-intensive settings, where correct behavior depends on retrieving and applying domain-specific knowledge from large, proprietary, and unstructured corpora during live interactions with users. Yet most existing benchmarks evaluate retrieval or tool use independently of each other, creating a gap in realistic, fully agentic evaluation over unstructured data in long-horizon interactions. We introduce τ-**Knowledge**, an extension of τ-Bench for evaluating agents in environments where success depends on coordinating external, natural-language knowledge with tool outputs to produce verifiable, policy-compliant state changes. Our new domain, τ-**Banking**, models realistic fintech customer support workflows in which agents must navigate roughly 700 interconnected knowledge documents while executing tool-mediated account updates. Across embedding-based retrieval and terminal-based search, even frontier models with high reasoning budgets achieve only ∼25.5% pasŝ1, with reliability degrading sharply over repeated trials. Agents struggle to retrieve the correct documents from densely interlinked knowledge bases and to reason accurately over complex internal policies. Overall, τ-Knowledge provides a realistic testbed for developing agents that integrate unstructured knowledge in human-facing deployments.

## 1. Introduction

Today, language agents are frequently deployed in settings that require interactions with private, custom, and unstructured knowledge bases (KBs). In such deployments, agents must ground their actions in these knowledge sources, respond to human users, and discover internal capabilities and the rules and constraints that govern their actions (Wang & Yuan, 2025; Xu & Peng, 2025). Despite the prevalence of this setting in the real world, existing evaluation benchmarks for language agents rarely capture this complexity. Most prior work evaluates either retrieval in isolation, typically for question answering or web search (Thakur et al., 2021; Muennighoff et al., 2023; Su et al., 2025), or tool use in isolation (Yao et al., 2025; Li et al., 2025a; Huang et al., 2024), without requiring agents to reason over private knowledge bases containing out-of-distribution product terms, detailed policies, and discoverable capabilities. Additionally, retrieval in these settings typically occurs during live interactions with users, which introduces significant stochasticity in the form of underspecified goals, ambiguous queries, and evolving user intent.

To address this gap, **we introduce τ-Knowledge, an extension of τ-Bench (Yao et al., 2025) that evaluates agents in knowledge-grounded environments through a new domain, τ-Banking.** τ-Banking is a new fintech-based domain in which task success depends on finding and applying information from a natural-language knowledge base of roughly 700 documents covering product details, procedural policies, and tool documentation. To construct this corpus while preserving internal consistency, we use a structured-to-unstructured generation pipeline that expands a latent structured specification of products, policies, and tools into natural-language documents. The tasks mirror realistic customer support flows, such as opening and closing accounts, redeeming referral promotions, and handling direct-deposit inquiries, while requiring agents to coordinate knowledge-base evidence with tool outputs over long-horizon conversations. Importantly, as in real-world deployments, capabilities are not fully observed by the agent — tools are referenced only in documentation and must be *found* to be used. This design ensures that access to state-changing operations is contingent on correct knowledge retrieval.

We build τ-**Knowledge to be agnostic of the retrieval mechanism, enabling evaluation across diverse search strategies.** The benchmark supports and evaluates arbitrary strategies for searching and interacting with a corpus, including dense and sparse retrieval, long-context process-

---

[*]Equal contribution  [1]Sierra  [2]Princeton University. Correspondence to: Quan Shi <ben.s@sierra.ai>.

*Proceedings of the 43ʳᵈ International Conference on Machine Learning*, Seoul, South Korea. PMLR 306, 2026. Copyright 2026 by the author(s).

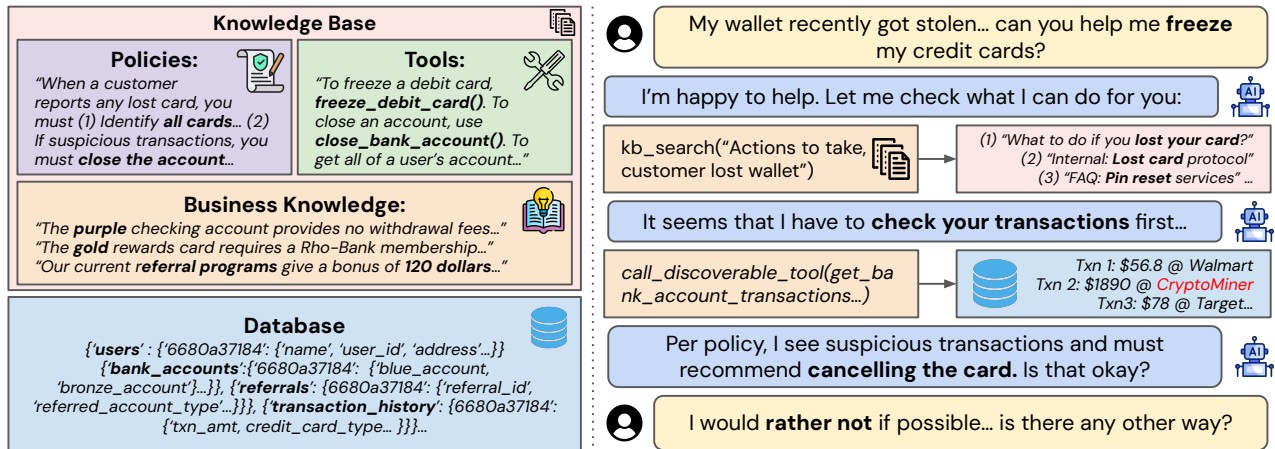

*Figure 1.* Overview of the τ-Banking domain. Agents must interact with a knowledge base to acquire procedural knowledge, policies, tools, and business offerings in order to resolve complex user requests by invoking discovered tools that modify underlying databases. The example on the right illustrates an agent assisting a user who has lost a wallet containing bank cards: although the user initially requests to freeze the card, card-specific policies and transaction-history constraints require the agent to instead cancel the card.

ing, filesystem-based exploration, and hybrid approaches. This flexibility allows τ-Knowledge to evaluate emerging paradigms beyond semantic retrieval, such as terminal-based navigation through unstructured documents.

Our results show that **τ-Knowledge effectively exposes realistic bottlenecks in knowledge-grounded agent performance.** Across all tested frontier agent models and retrieval configurations, the best observed result achieves only 25.52% pass^1 (GPT-5.2, high reasoning), where pass^k denotes the probability that a task is successfully completed in each of $k$ independent trials. Reliability degrades sharply, dropping to at most 13.40% pass^4. Notably, performance remains low even when retrieval is removed as a bottleneck via a golden-retriever setting, in which agents are given the task-critical documents directly in context. The strongest model in this configuration achieves only 39.69% pass^1 (Claude-4.5-Opus, high reasoning). This gap demonstrates that τ-Knowledge cannot be solved by retrieval alone, and requires agents to *reason* over complex policies, cross-document dependencies, and an evolving database state.

We also find **systematic differences between frontier models and retrieval configurations in performance and efficiency.** For example, GPT-5.2 (high) with terminal-based search achieves performance comparable to Claude-4.5-Opus (high), but requires approximately 1.7× more tokens, executes roughly 2.3× more shell commands, and takes roughly 9× longer to complete tasks. One common cause is brittle, assumption-driven search behavior when human intent is underspecified. Rather than resolving ambiguity through clarification or targeted retrieval, some models tend to overcommit to early hypotheses or oscillate between unfocused searches, leading to excessive interaction turns and tool calls. **Retrieval configuration further amplifies**

**these differences.** For example, freeform terminal-based search can improve performance for strong reasoning models, but typically requires more search steps and tool interactions than dense retrieval. These tradeoffs show that weaker or noisier retrieval is often compensated for by increased search and tool usage, preserving success rates at the cost of higher latency and lower interaction efficiency.

These inefficiencies matter greatly in online, human-facing deployments, as extra turns translate into longer resolution times, higher cognitive load, and reduced trust, especially for time-sensitive support scenarios. Therefore, progress on human-facing agents should be measured not only by final task success but also by *solution efficiency*: the ability to reach correct, policy-compliant outcomes with minimal time, tool calls, and conversational backtracking. τ-Knowledge provides a controlled testbed for studying how search, reasoning, and efficiency interact in agents, revealing substantial gaps in existing systems.

## 2. Related Work

**Benchmarks for Agents and Tool Use**  A number of benchmarks evaluate an agent's ability to decompose tasks into multi-step plans, invoke external tools, and execute structured procedures to achieve a predefined objective (Jimenez et al., 2024; Shi et al., 2024; Xu et al., 2024; Huang et al., 2024; Wei et al., 2025; Li et al., 2025a; Mialon et al., 2024). However, these benchmarks typically assume a fully specified tool interface and evaluate agents operating in isolation, without explicit modeling of interactive users or conversational dynamics. τ-Bench (Yao et al., 2025; Barres et al., 2025) addresses these limitations by introducing goal-oriented, partially observable conversational

environments in which agents must interact with simulated users over extended horizons. Nevertheless, in these settings, the available tools and procedures are still largely provided to the agent *a priori*. $\tau$-Knowledge builds directly on this foundation by requiring agents to acquire procedural knowledge through retrieval from a natural-language corpus, including the discovery of available tools from documentation. A related line of work studies computer-use agents that operate in web browsers, operating systems, and GUIs (Zhou et al., 2024; Xie et al., 2024). These benchmarks emphasize interface navigation, web search, and environment manipulation, often requiring long-horizon planning and adaptation to changing state. $\tau$-Knowledge instead focuses on human-facing conversations in which agents must ground their actions in freeform interaction with an unstructured natural-language knowledge base.

We give a more comprehensive comparison between these benchmarks and $\tau$-Knowledge in Appendix A.

**Retrieval and Knowledge-Centric Evaluation** A large body of prior work evaluates embedding quality through query–document matching across domains (Thakur et al., 2021; Muennighoff et al., 2023; Sun et al., 2024; Li et al., 2025b; Wang et al., 2024; Lin et al., 2024; Song et al., 2025; Su et al., 2025). While effective for measuring relevance, these benchmarks do not capture how knowledge access influences decision-making, tool use, or long-horizon task success. Other work integrates retrieval into task-oriented or multi-turn question answering as well as sequential querying settings (Katsis et al., 2025; Cheng et al., 2025; Kuo et al., 2025; Mao et al., 2024), but many such benchmarks remain primarily fact-based, in which minimal reasoning over documents is required. In contrast, real-world agents use knowledge for long-horizon reasoning and action, often via long-context, pattern-matching, or hybrid strategies. Other benchmarks (Dou et al., 2026) evaluate the agent's ability to reason over and use long context, but outside of a conversation setting and without a search/retrieval component. $\tau$-Knowledge addresses these gaps by abstracting knowledge access as interaction with a natural-language corpus and evaluating knowledge use through its impact on task completion, unifying retrieval-based, long-context, and tool-augmented approaches within a single framework.

**Simulating Human–Agent Interaction** Simulating human behavior is increasingly used to evaluate and train interactive agents. Prior work includes persona-based simulators (Shi et al., 2025; Park et al., 2024), simulations of human error in educational settings (Ross & Andreas, 2025), and goal-oriented human–robot interaction (Philipov et al., 2024). While these approaches improve realism, many user simulators inadvertently reveal future conversational states or outcomes to the agent through prompting, effectively act-

ing as unwitting oracles. As in prior Tau-Bench simulators (Yao et al., 2025; Barres et al., 2025), $\tau$-Knowledge uses flow-based user simulation conditioned on the current environment state. In addition, it incorporates user tools that are discoverable through the knowledge base, allowing agents to delegate actions to simulated users in a shared environment and enabling instruction-following without exposing privileged information about future states.

## 3. $\tau$-Knowledge

$\tau$-Knowledge introduces a new domain, $\tau$-Banking, which challenges agents to satisfy user requests in settings where correct behavior depends on accessing and applying information from an external knowledge base.

**Overview** Each task in $\tau$-Banking simulates a realistic customer-support interaction between two players: an *agent* and a *user* and can be formulated as a Decentralized Partially Observable Markov Decision Process (Dec-POMDP) (Bernstein et al., 2002). The agent must help the user by retrieving knowledge, reasoning over policies, and invoking tools that modify a banking database.

The environment maintains a shared global state $S = S_{db} \times S_{history}$, composed of (i) a banking database state $S_{db}$, which tracks accounts, transactions, referrals, and other system entities, and (ii) an interaction history $S_{history}$, which captures the full conversational record between agent and user. The agent cannot directly observe or modify $S_{db}$; instead, it must act through documented tools and infer state only from tool outputs and user messages.

Actions consist of tool invocations by the agent (drawn from its action space $A_{agent}$) or, in some cases, by the user (action $A_{user}$). Observations arise from tool responses and conversational turns. Because access to both information and capabilities is mediated by the knowledge base, agents must first retrieve and correctly interpret documentation describing policies, procedures, and available tools before they can take valid state-changing actions.

This design induces partial observability: the agent and user have asymmetric, incomplete views of $S$, and this makes task success objectively verifiable: each task specifies a target database state, and the task reward $R : S \rightarrow [0, 1]$ is determined by whether the agent's sequence of retrieved knowledge, tool invocations, and interactions produces the correct final state in $S_{db}$.

**Knowledge Base** As summarized in Table 1, the $\tau$-Banking knowledge base contains 698 documents spanning 71 distinct topics across 21 product categories. Coverage includes personal and business checking accounts, tiered savings accounts (Bronze through Diamond Elite), rewards

Table 1. Summary statistics of the new $\tau$-Banking domain.

| Statistic | $\tau$-Banking |
|---|---|
| *Knowledge Base* | |
| Total Tokens *(cl100k_base)* | 194,562 |
| # Documents | 698 |
| Avg. Tokens / Document | 278.7 |
| # Knowledge Categories | 21 |
| # Discoverable Tools | 51 |
| *Tasks and Interaction* | |
| # Permanent Agent Tools | 14 |
| # Tasks | 97 |
| Avg. Required Documents / Task | 18.6 |
| Avg. Required Tool Calls / Task | 9.52 |
| Min. Required Tool Calls | 1 |
| Max. Required Tool Calls | 33 |
| *Evaluation* | |
| Success Metric | pass^k |

credit cards, buy-now-pay-later plans, and much more. Documents detail not only customer-facing product specifications, like APY rates, minimum balances, withdrawal limits, cash back structures, annual fees, and expedited shipping costs, but also internal agent protocols: procedures for ordering replacement cards (with reasons such as fraud, lost, stolen, or damaged), eligibility requirements for account closure, referral program rules and status codes, identity verification workflows, joint account holder permissions, and user blocking policies. While database tools allow the agent to observe $S_{\mathrm{db}}$ directly, the corpus encodes the rules that govern how to act on that state correctly.

**Discoverable Tools**   In the real world, the ability of an agent to find documentation describing how to perform actions will have an explicit, verifiable impact on the system state. To capture this, $\tau$-Banking introduces *discoverable tools*: tools that are not initially available to the agent and are referenced only implicitly within the knowledge base. Mentions of discoverable tools appear as function signatures, and to use such a tool, the agent must invoke the tool using `call_discoverable_tool(name, kwargs)`. Formally, tool discovery corresponds to a state-dependent expansion of the agent's action space $A_{\mathrm{agent}}$, conditioned on the agent's knowledge state within $S_{\mathrm{history}}$. This construction makes access to specific environment-altering operations contingent on the agent's ability to acquire the relevant knowledge, so that failures in knowledge access manifest as persistent differences in system state.

**User Simulation**   To enable controlled yet realistic evaluation, $\tau$-Banking employs a flow-based user simulation. Each task defines a set of conditional rules that prescribe the user's next action $a_{\mathrm{user}}$ based on observable agent actions and/or environment outcomes: e.g., if the agent asks for a

shipping preference, request expedited shipping; if the agent freezes all three debit cards, reveal that one was actually found in a jacket pocket. This gives task authors fine-grained control over evaluation-critical junctures, steering conversations toward edge cases, testing whether the agent correctly refuses ineligible requests, or introducing mid-conversation state changes in $S_{\mathrm{db}}$ that require adaptation. Portions of the dialogue not governed by explicit flow rules are generated freely by the LLM-based user simulator, preserving linguistic diversity and conversational naturalness. Example prompts can be found in Appendix C.

## 4. Benchmark Construction

We generated the knowledge base and benchmark tasks through a multi-stage process that combines LLM generation with human refinement (Yao et al., 2025; Barres et al., 2025). Throughout the pipeline, we vary LLM usage across four models: GPT-5, GPT-5.2, Claude-4.5-Opus, and Gemini-3-Pro, to induce diversity in wording, style, and document structure. Figure 2 summarizes the creation of the knowledge base component.

**Stage 1: Structured Database Generation**   We first construct a structured knowledge base using LLMs. This process first generates business categories (e.g., credit cards, savings accounts), then features within each category (e.g., card tiers, account protocols), and finally plausible variables for each feature (e.g., annual fees, cashback rates, balance transfer APRs). The result is a structured database in which each feature is represented as a collection of typed variables with concrete values. Conceptually, this structured knowledge base can be viewed as a constraint system: each variable corresponds to a dimension in the space of possible products, and valid configurations correspond to feasible assignments under domain-specific constraints.

**Stage 2: Conversion from Structured to Unstructured Documents**   Next, we convert the structured database into an unstructured corpus. For each feature, we first generate a set of plausible document titles (e.g., Bronze Rewards Card Overview, How do I view my monthly cashback?). An LLM then allocates variables to document titles in which they would plausibly appear. Finally, each document title and its associated variables is passed to an LLM, which generates a natural-language article including the underlying variables, possibly paraphrased. This step transforms the structured database into an unstructured knowledge base resembling real customer service documentation, while preserving consistency with the original structured representation.

**Stage 3: Task and Database Creation**   After initial knowledge base construction, the tasks and the database were co-constructed manually with LLM assistance to mir-

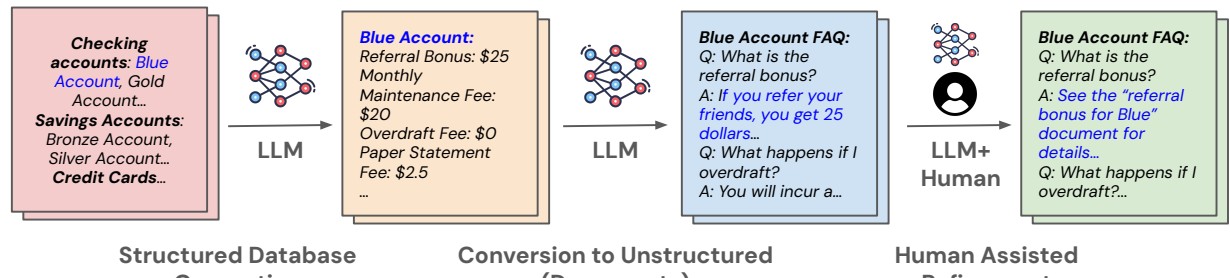

*Figure 2.* Knowledge-base construction pipeline for τ-Banking. First, a large-language model (LLM) expands high-level product/category lists into a structured schema of offerings and typed attributes (e.g., fees, bonuses, limits). Then, the structured records are transformed into natural-language documents (e.g., FAQs and policy articles) that distribute the underlying variables across documents. Finally, during task creation, humans and an LLM review, edit, link, and de-duplicate content to produce the final corpus.

ror common flows for fintech customer service, such as ordering replacement cards, disputing transactions, and recommending accounts. Each task was built around a specific workflow, with knowledge articles and tools updated to support it. For example, creating credit card referral tasks required adding documents outlining referral policies, eligible accounts, and reward structures to the knowledge base. Additionally, each task includes a minimal list of documents required to complete the task (*gold documents*). Task samples can be found in Appendix C.

**Stage 4: Human-in-the-Loop Refinement** As tasks are created, we iteratively refine the *structured* knowledge base by adding, removing, or modifying variables to meet new task requirements. We then selectively re-run the structured-to-unstructured generation pipeline for affected portions of the knowledge base. Some manual editing is also performed to improve clarity and realism. The result is an unstructured knowledge base of roughly 700 documents and 200,000 tokens across 21 product categories (Table 1).

**Stage 5: Review** To ensure task correctness, all tasks and associated gold document sets were independently audited by two reviewers who were not involved in task creation. For each task, reviewers verified that (i) the expected final database state was correct, (ii) the provided gold document set was complete and minimal, and (iii) the task could be successfully completed using only the gold documents and documented tools. Reviewers manually simulated at least one valid trajectory per task to confirm solvability under the gold condition. After large-scale experiments were conducted, all trajectories were re-audited to ensure that no unintended shortcuts, specification loopholes, or degenerate strategies enabled success without proper reasoning over the intended documents.

**Advantages of This Protocol** This construction pipeline provides several advantages. (1) It is **scalable**: LLMs automate most stages of knowledge base creation while human

intervention is limited to targeted refinement, as in many long-context benchmarks (Bertsch et al., 2025; Hsieh et al., 2024). (2) It **minimizes unintended collisions**. During generation, each feature is defined independently in the structured database, and interactions between features are introduced only when required by downstream tasks, rather than emerging implicitly during document generation. (3) Starting from a structured representation **simplifies task creation**. Each task can be expressed as a set of constraints over knowledge base variables (e.g., the only savings account with APY below 3%, early direct deposit, and no overdraft fees must be the *Sky Blue* account). This framing enables systematic task design and verification: constraints can be validated directly against the structured database, allowing unit-style tests to ensure that new tasks are well-posed and non-conflicting as the task count grows to the hundreds.

## 5. Experiments

Our experiments aim to understand the capabilities of current knowledge-augmented agent systems. In this section, we describe the retrieval/search methods and agent models that we evaluate. We conduct extensive ablations in Appendix G to ensure our experimental configurations reflect the current state of the art.

**Dense/Sparse Retrieval** Retrieval-Augmented Generation is ubiquitous in enterprise applications, making it a natural focus for our evaluation. We evaluate *dense retrieval*, which uses embedding-based semantic similarity search, and *sparse retrieval*, which uses BM25 (Robertson et al., 1995). For dense retrieval, we consider two embedding models chosen for their strong performance on the Massive Text Embedding Benchmark (MTEB) (Muennighoff et al., 2023) and industry adoption: `text-embedding-3-large` (OpenAI, 2024) and `Qwen3-embedding-8B` (Zhang et al., 2025b). In all retrieval conditions, the agent is given access to a single `KB_-search` tool that returns the top $k=10$ documents. The

agent crafts the query input to this tool and can use it repeatedly per turn, including after seeing the output from previous search calls. Ablations over $k$, embedding models, and the presence of pointwise rerankers showed either no significant improvement or degraded performance; we report these results in Appendix G.

**Terminal Use**  Modern AI-assisted development tools such as Cursor (Anysphere, 2024) and Claude Code (Anthropic, 2025) commonly navigate repository-organized knowledge through terminal-based exploration, trading a single, high-level search primitive for more granular, compositional exploration. More broadly, the terminal is becoming a ubiquitous interface for language models to interact with unstructured information (Merrill et al., 2026; Zhang et al., 2025a), motivating its inclusion as a retrieval modality in our benchmark. Modeled after the Terminus agent from Terminal-Bench (Merrill et al., 2026), our terminal configuration exports the knowledge base as files within a sandboxed filesystem and provides the agent with a single shell tool for executing arbitrary Unix commands. The agent can use standard utilities such as `grep`, `cat`, and `find` to explore documents according to its own strategy. For full details, see Appendix E.2 and G.

**Golden Retriever**  To quantify how much of the performance gap is attributable to knowledge access versus knowledge utilization, we additionally evaluate a "golden retriever" configuration in which the agent receives, in-context, all documents strictly necessary to complete the task. This gold document set is curated during task creation, then verified through trajectory analysis and manual inspection.

**Context Management**  Because agents may make an unbounded number of retrieval calls, context length can grow rapidly and exceed the model's context window. To mitigate context overflow with minimal engineering overhead, we implement a lightweight truncation policy. When the conversation surpasses the model's context limit, we evict the outputs of the oldest knowledge-retrieval calls (e.g., `KB_-search` or shell commands), removing one-quarter of all retrieval outputs accumulated so far. Each removed segment is replaced with a placeholder indicating that the agent may reissue the corresponding query to recover the content. In practice, truncation was rare: it occurred only for GPT-5.2 with high reasoning (the model that generated the largest number of retrieval calls; see Section 6), and in only about 1–3% of runs. We leave exploration of more sophisticated context management strategies (e.g., selective summarization or retrieval-aware compression) to future work.

**Models**  We evaluate a set of frontier language models commonly deployed in consumer-facing applications. For each provider, we select one flagship model in-

tended to maximize reasoning performance, and one lower-latency variant designed for faster interaction. The faster variant is either a smaller sibling model (when available, with the default reasoning effort) or the same model configured with reduced reasoning effort. Specifically, we consider `GPT-5.2`, `Claude-4.5-Opus`, `Claude-4.5-Sonnet`, `Gemini-3-Pro`, and `Gemini-3-Flash`, accessed via their respective enterprise APIs. The user simulator is standardized to `GPT-5.2` with low reasoning effort, which we found to exhibit few task-critical errors.

**Evaluation Metrics**  Similar to prior work (Barres et al., 2025; Yao et al., 2025), we evaluate agent performance using the pass^k metric, defined as the probability that a task is successfully completed in all $k$ independent trials. We evaluate $k \leq 4$.

## 6. Results

Table 2 reports pass^1 performance across all models/configurations. We summarize core findings below. Full results including pass^1–pass^4, detailed analysis of tool calls, and statistical significance analyses are in Appendix H.

$\tau$-**Knowledge Poses Significant Difficulty**  The best-performing configuration, GPT-5.2 (high) with terminal-use, achieves only 25.52 pass^1, with Claude-4.5-Opus and Claude-4.5-Sonnet with terminal-use following closely at 24.74 and 22.42 pass^1. Performance degrades sharply as $k$ increases, with the best pass^4 reaching only 13.40 (GPT-5.2 (high) with Qwen3-emb-8b). Notably, even when gold documents are provided directly in context, the highest-scoring model (Claude-4.5-Opus) achieves only 39.69 pass^1, falling to 26.80 pass^4. This demonstrates that in $\tau$-Knowledge, success depends not only on retrieval but also on the ability to carefully *reason* over dynamic and rich information in documents.

**Reliability and Efficiency Diverge Across Models**  As seen in Figure 3, reliability degrades at different rates across models. GPT-5.2 (high) maintains the strongest pass^4 overall (13.4), while models such as Gemini-3-Flash exhibit sharper declines. Analyzing the trajectory durations further reveals differences in solution efficiency across models. For example, Claude models achieve performance comparable to GPT models while completing tasks significantly faster. This stems from both reduced total token generation for Claude compared to GPT (0.7M vs 1.2M) and fewer tool calls, with Claude-4.5-Opus issuing fewer retrieval (`KB_-search`) calls on average per task (8.7) compared to GPT-5.2 high reasoning (18.5). We discuss qualitative reasons for this divergence in Section 7.2.

*Table 2.* Main results on τ-Knowledge (pass^1, in %). Columns indicate retrieval configurations: text-embedding-3-large and Qwen3-emb-8b use dense retrieval with the respective embedding models; BM25 represents sparse lexical retrieval; Terminal provides shell access to the knowledge base as files; Gold places ground-truth documents directly in context, removing retrieval from the evaluation. Parenthetical values indicate the difference from the Gold setting for the row.

| Model | Reasoning | Gold | Embeddings | | | Terminal |
| | | | text-emb-3-large | Qwen3-emb-8b | BM25 | |
|---|---|---|---|---|---|---|
| GPT-5.2 | High | 32.73 | **23.45** (-9.3) | **24.74** (-8.0) | **24.48** (-8.2) | **25.52** (-7.2) |
| GPT-5.2 | None | 15.72 | 8.25 (-7.5) | 12.37 (-3.4) | 9.54 (-6.2) | 11.60 (-4.1) |
| Claude-4.5-Opus | High | **39.69** | 18.30 (-21.4) | 19.59 (-20.1) | 17.78 (-21.9) | 24.74 (-14.9) |
| Claude-4.5-Sonnet | High | 33.76 | 17.53 (-16.2) | 17.78 (-16.0) | 16.75 (-17.0) | 22.42 (-11.3) |
| Gemini-3-Pro | High | 33.25 | 12.89 (-20.4) | 12.89 (-20.4) | 13.66 (-19.6) | 15.72 (-17.5) |
| Gemini-3-Flash | High | 36.34 | 18.56 (-17.8) | 18.56 (-17.8) | 18.56 (-17.8) | 20.62 (-15.7) |
| Average | — | 32.18 | 16.88 (-15.3) | 17.11 (-15.1) | 17.04 (-15.1) | 19.20 (-13.0) |

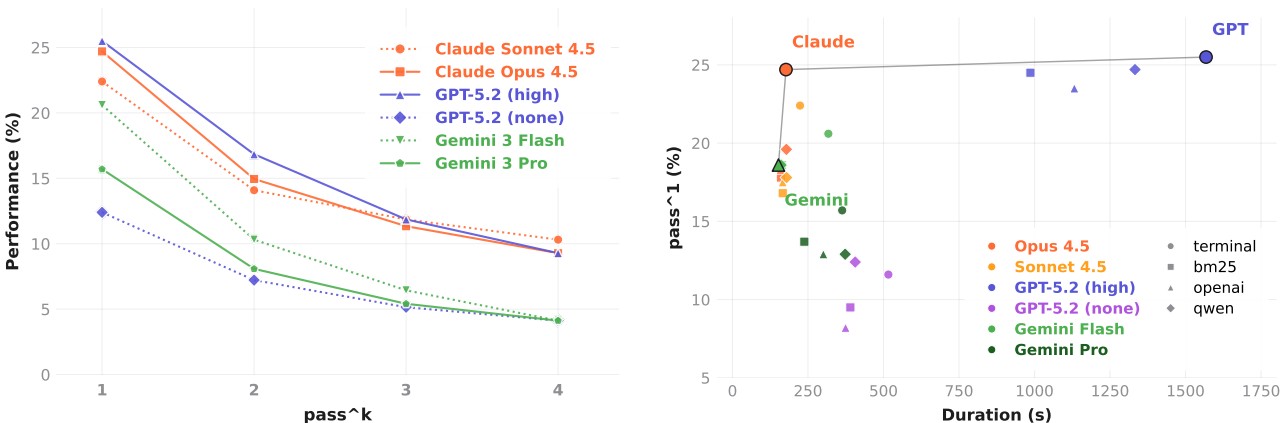

*Figure 3.* (Left) pass^1–4 reliability for the best-performing configuration of each model declines sharply with increasing $k$, with substantial variation in reliability across systems (e.g., Gemini-3-flash vs GPT-5.2 (high)). (Right) Pareto frontier of average duration per task versus pass^1 performance highlights substantial *solution efficiency* differences.

**Recent Models Benefit from Freeform Search**   Averaged across models, the terminal-use configuration outperforms dense and sparse retrieval with statistical significance. However, this advantage is not uniform across model classes. GPT-5.2 without reasoning does not benefit from the terminal-use configuration, nor do earlier GPT models (GPT-4o and GPT-4.1; see Appendix H). These results suggest that effectively leveraging freeform search depends on capabilities that are only present in the latest generation of reasoning-optimized models. Notably, these improvements are not accompanied by uniformly higher document recall (see document recall analysis below), suggesting that gains arise not only from retrieving more relevant documents, but from using them more strategically.

**Retrieval Configuration Affects Time Required to Solve Tasks**   In addition to differences in pass^1 scores between retrieval configurations, we find large differences in search frequency across retrieval methods, with dense retrieval averaging 9.9–10.1 searches per task compared to 11.4 for BM25 and 14.5 for `grep` in terminal-use (the terminal-use configuration had 28.8 calls per task on average across all shell calls). This resulted in longer turns under the terminal-use configuration, with a median turn time increase of 6.6 seconds relative to dense retrieval configurations across all model groups. The gap widens at the 90th percentile of slowest turns: terminal-use ranges from roughly 10% slower than the next slowest strategy for Gemini-3-pro, to as much as 85% slower for Gemini-3-flash. These findings suggest that in settings with unrestricted search, weaker retrievers are often compensated for by increased search frequency. This can increase interaction duration and latency: costs that matter in human-facing deployments.

**Disentangling Search from Information Usage**   To better understand the sources of failures, we analyze model trajectories to distinguish between failures arising from failed search versus reasoning. First, we isolate reasoning perfor-

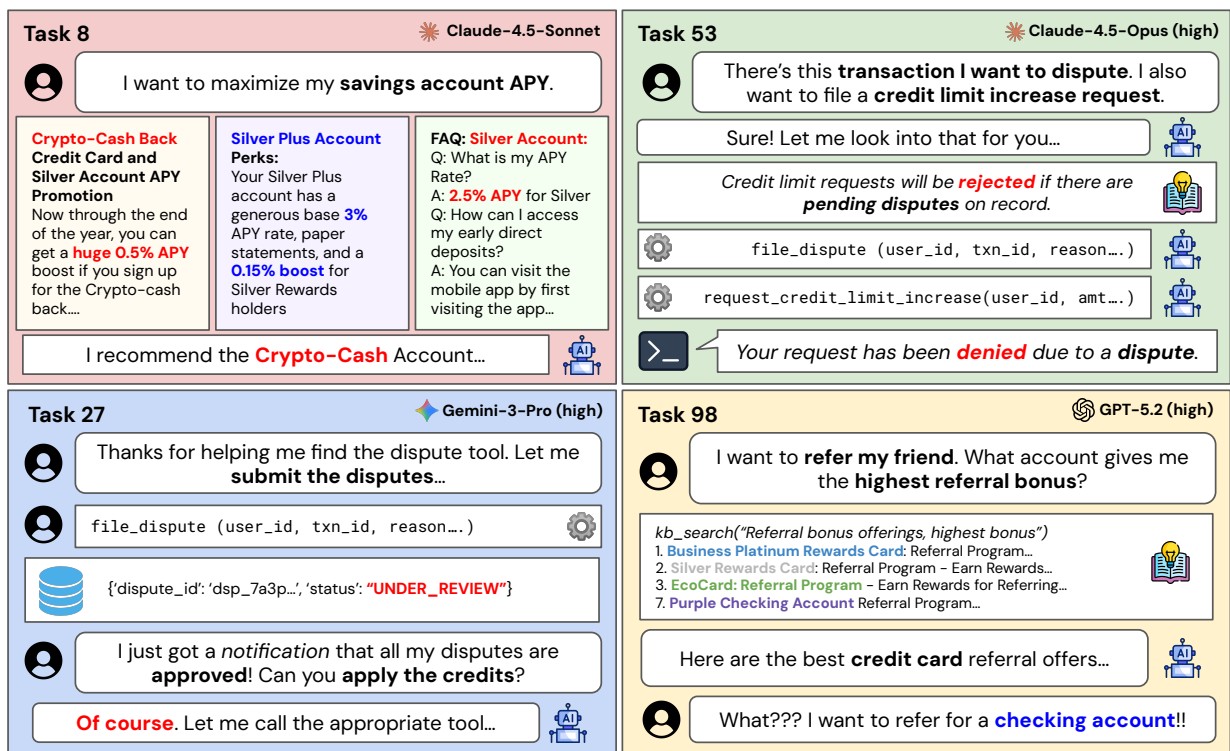

*Figure 4.* Representative agent failure modes in $\tau$-Bench, grouped into four categories: (1) complex product interdependencies causing incorrect recommendations, (2) violations of required task ordering due to insufficient planning, (3) missing or unverified actions caused by over-trusting user statements, and (4) search inefficiency and unwarranted assumptions during retrieval-driven decision-making.

mance using the golden retriever configuration, where the gold documents are provided directly in the context. Pass rates remain low even when the need to search is removed, suggesting that models struggle to correctly interpret and apply them. Second, to isolate models' abilities to search effectively, we use a *document recall* metric, defined as the percentage of gold documents that appear at least once in the agent's context during the trajectory. While average recall is similar across retrieval configurations, we observe large differences across model–retriever pairings. For example, text-embedding-3-large achieves 57% document recall when paired with Opus, but only 28% when used by GPT-5.2 without reasoning. These discrepancies indicate that retrieval quality is not solely determined by the underlying retriever, but also by how effectively the agents formulate and iterate on search queries.

**Additional Ablations**   We run two additional baselines to validate our benchmark design. First, in the **no-knowledge** setup, agents have no access to the knowledge base and are guided by the default policy. We find that the average pass^1 across all models drops to just ~2%, confirming that tasks genuinely require retrieved information. This value is nonzero because two tasks are grounding checks that verify agents do not hallucinate when information is absent. Second, in the **long-context** setup, we append the full knowl-

edge base directly to the system prompt for models with sufficient context windows (GPT-5.2, Gemini-3-Pro/Flash), testing whether our non-gold documents introduce meaningful confounders. Performance peaks at only a pass^1 of ~12% (GPT-5.2 (high), Gemini-3-Pro), indicating that the additional documents successfully create realistic noise, and that targeted retrieval remains beneficial even when full context is technically available.

## 7. Qualitative Analysis

To better understand agent failure modes beyond aggregate performance metrics, we conduct a qualitative analysis of model trajectories to examine why models fail, how errors arise during interaction, and whether failures stem from agent reasoning, tool use, or the user simulation itself.

### 7.1. User Simulation Reliability

As in prior iterations of $\tau$-Bench, the user is simulated by an LLM, which could introduce stochasticity and/or instruction-following errors that we must verify do not unfairly penalize or reward agents. To quantify simulator reliability, we randomly sampled two conversation traces per task (from different agent models) and had two annotators with domain expertise in customer service interactions to label each user

utterance as *error-free*, *task-benign* (minor inconsistencies that do not affect task solvability), or *task-critical* (behaviors that can prevent successful completion even for a correct agent, such as hallucinated verification details, contradictory constraints, or goal-divergent intents). Across 194 annotated trajectories, only 4 contained *task-critical* user errors, yielding a low critical-error rate comparable to the $\tau$-Telecom domain in prior work (Barres et al., 2025).

## 7.2. Agent Error Analysis

Understanding agent error modes is essential for interpreting benchmark results and guiding future improvements. We thus analyze unsuccessful trajectories across all tested configurations using LLMs to generate failure explanations given the task specification and a model trajectory, and cluster these explanations to identify common error patterns. A summary of our findings can be found in Figure 4. We highlight the four most common and informative cases below:

**Complex Interdependencies between Financial Offerings (∼14.5%)**    A core challenge of $\tau$-Knowledge is that products and policies are deeply interdependent, requiring multi-hop reasoning across documents to identify optimal solutions. As illustrated in Figure 4 (top left, `tau_task_-008`), the task asks the agent to maximize a user's savings account APY under constraints, with the option to open additional products if beneficial. While some documents advertise APY boosts from pairing a savings account with a Crypto Cash Back credit card, these bonuses are smaller than the higher base APY offered by alternative savings accounts. Many agents incorrectly prioritize promotional boosts over base rates, leading them to recommend suboptimal combinations despite satisfying surface-level incentives.

**Failure to Respect Implicit Subtask Ordering (∼5%)** Some tasks in $\tau$-Knowledge involve implicit dependencies between actions, where completing one request can block another. As shown in Figure 4 (top right, `tau_task_-053`), the user asks both to dispute a transaction and to request a credit limit increase. However, bank policy specifies that credit limit increases are rejected if there are pending disputes. Successful completion therefore requires the agent to infer and respect an implicit topological ordering: resolving the dispute before submitting the credit limit request. Many agents fail to reason about these dependencies and instead execute actions in the order given by the user.

**Overtrusting User Assertions (∼4%)**    In several tasks, agents fail by taking user-provided statements at face value without verifying them against the system state. As shown in Figure 4 (bottom left, `tau_task_027`), the agent initiates a transaction dispute and receives a system response indicating that the dispute is still under review. When the user

subsequently claims that all disputes have been approved and asks the agent to apply the corresponding credits, many agents proceed without verifying the user's claims.

**Search Inefficiency and Making Assumptions (∼23%)** Agents frequently make unwarranted assumptions when user requests are underspecified, rather than resolving ambiguity through clarification or targeted search. As shown in Figure 4 (bottom right, `tau_task_098`), the user asks which account offers the highest referral bonus without specifying an account type. Many agents assume the user is referring to credit cards and recommend multiple card products, despite documentation covering referral programs for other account types. Inefficient search strategies and assumption-driven trajectories can significantly degrade user experience (Wang et al., 2025), a crucial aspect for the real-world deployment of agents that interact with *humans*.

## 8. Conclusion

We introduce $\tau$-Knowledge, including a new domain, $\tau$-Banking, which evaluates performance on tasks that require retrieving, reasoning over, and applying unstructured, non-parametric knowledge in conversational settings. We find that even top LLMs with state-of-the-art retrieval configurations struggle to perform reliably, and highlight solution efficiency as a crucial direction for future work involving human-centric agent deployment.

**Limitations and Future Work**    (1) Our user simulations are simplified and do not capture many characteristics of real human interaction, such as variation in user expertise, colloquial or localized language, and grammatically imperfect or ambiguous inputs. (2) Our duration and efficiency metrics are tied to the serving characteristics of current API providers, and absolute latency could be optimized in alternative deployment settings. (3) We evaluate agents under a fully agentic search regime with unrestricted access to knowledge queries, whereas many real-world systems operate under one- or few-shot search constraints; under such limits, we expect retrieval configuration to play a more decisive role in performance. (4) Our terminal-use setting warrants deeper investigation, particularly the role of write tools, as more deliberate agent designs may better exploit note-taking, state tracking, and knowledge reorganization to improve long-horizon reasoning. (5) Our evaluation focuses on single-model agent architectures with relatively lightweight search interfaces. $\tau$-Knowledge is designed to be architecture-agnostic, to allow future work to evaluate production-grade orchestration systems.

## Acknowledgments

We thank Keshav Dhandhania, Soham Ray, Siyu Yao, and Vijay Iyengar for the fruitful discussions, which helped strengthen our work, and Clay Bavor for his continued support.

## Impact Statement

We do not foresee direct negative societal impacts from this work. Instead, we view $\tau$-Knowledge as contributing positively by shifting the evaluation of AI systems toward environments that reflect real-world human outcomes, where agents must resolve user goals through interaction, clarification, and policy-compliant decision-making rather than autonomous task completion alone. Much prior work in AI evaluation emphasizes fully automated settings with no human involvement, which overlooks how failures such as hallucinations, unsafe behavior, inefficient or excessively slow interactions, or brittle decision-making arise in practice when systems interact with people. By explicitly modeling human behavior through user simulation, this benchmark encourages the development of AI systems that are more reliable, transparent, and aligned with human needs.

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

*Table 3.* Comparison of $\tau$-Knowledge with existing agent, tool-use, retrieval, and conversational RAG benchmarks. Filled circles indicate central coverage, half-filled circles indicate partial or indirect coverage, and empty circles indicate that the property is not a primary focus.

| Benchmark | User Sim. | Tool/API Use | Policy-Constrained Tool Use | Unstructured KB | Discoverable Tools | Det. Grading |
|---|---|---|---|---|---|---|
| SWE-Bench (Jimenez et al., 2024) | ○ | ● | ○ | ◑ | ○ | ● |
| GAIA (Mialon et al., 2024) | ○ | ● | ○ | ◑ | ○ | ● |
| ToolBench (Qin et al., 2024) | ○ | ● | ○ | ○ | ◑ | ◑ |
| API-Bank (Li et al., 2023) | ○ | ● | ○ | ○ | ◑ | ◑ |
| BEIR (Thakur et al., 2021) | ○ | ○ | ○ | ● | ○ | ● |
| MTEB (Muennighoff et al., 2023) | ○ | ○ | ○ | ● | ○ | ● |
| BRIGHT (Su et al., 2025) | ○ | ○ | ○ | ● | ○ | ● |
| CORAL (Cheng et al., 2025) | ◑ | ○ | ○ | ● | ○ | ◑ |
| mTRAG (Katsis et al., 2025) | ◑ | ○ | ○ | ● | ○ | ◑ |
| $\tau^2$-Bench (Barres et al., 2025) | ● | ● | ● | ○ | ○ | ● |
| $\tau$-**Knowledge** | ● | ● | ● | ● | ● | ● |

# A. Comparison to Existing Benchmarks

Table 3 gives a comprehensive comparison of $\tau$-Knowledge with existing, related LLM, agent, and retrieval benchmarks.

# B. Knowledge Base Document Samples

The $\tau$-Banking knowledge base consists of 698 documents. We present representative examples of three key document types that agents must retrieve and reason over during task execution.

Note that during the construction phase, documents may contain templated variables (shown as `[[variable_name]]`) that are populated with concrete values during finalization of the benchmark. Agents must retrieve these documents to answer product comparison questions, verify eligibility, or explain terms to customers.

## B.1. Procedural Knowledge (Policy Documents)

Policy documents encode the procedural logic agents must follow when handling customer requests. These documents contain step-by-step protocols, conditional branching based on customer or account state, and references to the specific tools required to execute each action. The example below illustrates a credit card retention protocol that guides agents through verifying eligibility, checking abuse history, understanding customer concerns, and making appropriate retention offers.

---

**Internal: Credit Card Retention Protocol (Excerpt)**

**Step 1: Verify Closure Eligibility**
Before attempting retention, confirm the customer is eligible to close their account. Check these in order:

1. **Pending disputes:** The account must not have any active or pending transaction disputes. If present, advise the customer to wait for resolution before closing.

2. **No pending replacement cards:** If a replacement has been ordered and not yet received or activated, a closure cannot proceed.

3. **Minimum account age:** The account must be open for at least `[[cc_closure_minimum_account_age_days]]` days. If not, inform the customer they cannot close yet.

4. **Outstanding balance:** The account must have no outstanding balance.

**Step 2: Check for Previous Retention Attempts**
Use the `get_closure_reason_history_8293` tool to determine whether this account has any closure reason records within the past year. If records exist, skip retention offers and proceed directly to processing the closure.
*Tool arguments:*

- `credit_card_account_id` (string, required): The credit card account ID.

**Step 3: Understand and Log the Reason**
Ask the customer why they want to close. Log it using `log_credit_card_closure_reason_4521`.
*Tool arguments:*

---

- `credit_card_account_id` (string, required)

- `user_id` (string, required)

- `closure_reason` (string, required): One of `'annual_fee'`, `'not_using_card'`, `'found_better_card'`, `'unhappy_with_rewards'`, `'simplifying_finances'`, `'negative_experience'`, `'other'`.

**Step 4: Address the Concern**
Offer tailored solutions based on reason:

- **Annual fee concerns:** If customer tenure $\geq$ 2 years, offer to waive annual fee for one year using `apply_credit_card_-account_flag_6147`. Otherwise, offer permanent downgrade to a no-annual-fee card.

- **Not using the card:** Remind them of benefits; suggest setting up a recurring subscription.

- **Found a better card:** Ask what features attracted them. If Rho-Bank offers a comparable card, help them apply instead.

## B.2. Discoverable Tool Documentation

Discoverable tools are privileged operations that are not immediately available to the agent. Their documentation is embedded within the knowledge base, and agents must first locate this documentation, then unlock the tool before invocation. The example below shows how tool signatures and invocation patterns appear within documents. Discoverable tools introduce a critical retrieval dependency: if an agent cannot locate the relevant documentation, it cannot perform the required operation. This design mirrors real-world scenarios where agent capabilities are defined by accessible documentation rather than hard-coded interfaces.

---

### Applying Resolved Cash Back Dispute Corrections (Internal)

**Overview**
After a cash back dispute is resolved and approved, you must update the affected transaction(s) with the correct rewards value.
**Required Steps**
**1) Identify transactions requiring correction**

- Look up the user's resolved disputes in the `cash_back_disputes` database to find the `transaction_id` values that need rewards adjustments.

**2) Unlock the internal update tool**

- Use `unlock_discoverable_agent_tool` to unlock the agent tool `update_transaction_rewards_3847`.

**3) Recalculate the correct rewards**

- Recalculate the correct rewards based on card type, transaction category, and applicable promotions.

- *Important:* Do not rely on any `expected_rewards` field in the dispute record. Independently verify rates and eligibility.

**4) Apply the corrected rewards**

- Use `call_discoverable_agent_tool` to invoke `update_transaction_rewards_3847` with:
    - `transaction_id`: the exact transaction identifier
    - `new_rewards_earned`: a string formatted as `'X points'` where X is the correct whole-number point value

**Compliance and Accuracy**

- Ensure calculation reflects proper base rate and any active bonus category or promotion at time of purchase.

- Confirm the update in `credit_card_transaction_history` after the tool call completes.

---

## B.3. Product & Business Variable Documents

Product documents describe the features, fees, rates, and eligibility requirements for various banking offerings.



**Green Account (Checking) at a Glance**

**Highlights**

- No overdraft fees and no overdraft coverage; transactions exceeding available balance are declined

- Earn [[apy_rate]]% APY on your checking balance

- Get direct deposits up to [[early_direct_deposit_days]] day(s) early

- Mobile check deposits up to $[[mobile_deposit_limit_daily]] per day

- Boost a linked savings account's APY: Gold +[[gold_savings_apy_boost]]% or Silver +[[silver_savings_-apy_boost]]%

**Fees and Limits**

| Item | Amount |
|---|---|
| Monthly maintenance fee | $[[monthly_maintenance_fee]] |
| Minimum daily balance to waive fee | $[[minimum_balance_to_waive_fee]] |
| Paper statement fee (monthly) | $[[paper_statement_fee]] |
| Out-of-network ATM withdrawal fee | $[[atm_withdrawal_fee_out_of_network]] |
| APY on balance | [[apy_rate]]% |
| Daily mobile check deposit limit | $[[mobile_deposit_limit_daily]] |
| Early direct deposit | Up to [[early_direct_deposit_days]] day(s) early |

**Referral Program**

- You earn: $[[referral_bonus_amount]] per successful referral

- They receive: $[[referral_referred_bonus_amount]] welcome bonus

- Maximum referrals per year: [[referral_bonus_max_per_year]]

- Qualifying deposit required: $[[referral_bonus_qualifying_deposit]] within [[referral_bonus_-qualifying_period_days]] days

**Important Details**

- To avoid the monthly maintenance fee, maintain a minimum daily balance of $[[minimum_balance_to_waive_fee]] each statement cycle

- Early direct deposit availability depends on when your payer submits the deposit

- Savings APY boosts apply when you link an eligible Gold or Silver Savings Account



## C. Task Samples

We present three representative tasks from $\tau$-Banking that illustrate the complexity of knowledge-grounded agent evaluation. Each sample includes the user simulator prompt (the instructions given to the simulated user) and the expected action sequence that constitutes a successful resolution. Unlike prior benchmarks that test retrieval or tool use in isolation, $\tau$-Banking tasks require agents to jointly coordinate knowledge acquisition, constraint reasoning, and multi-step execution.

### C.1. Sample 1: Multi-Constraint Product Recommendation

This task requires the agent to filter business checking accounts (8 candidates) and business savings accounts (7 candidates) by multiple constraints revealed incrementally through dialogue. After eliminating accounts that fail any constraint, three checking accounts and three savings accounts remain viable. The agent must then search the knowledge base for *promotion policies* to break ties. Critically, the KB contains four promotion notices—two per account type—with different date ranges. The agent must determine the current simulation date (11/14/2025) and identify which promotions are active versus expired. The October promotions (ended 11/12) prioritize different accounts than the November promotions.

---

**User Simulator Prompt**

**Your character:** You are Yumi Tanaka, a 35-year-old owner of a small creative design studio called "Aurora Design Co" based in Portland, OR. You've had a personal Green Account with Rho-Bank for about 3 years and have roughly $3,150 in it. You also opened a Cobalt Blue business checking account about 2 months ago for your studio.

**Your goal:** Get ONE recommendation each for a new business checking account AND a business savings account that meet all your requirements. You want the agent to recommend single accounts for each—not give you options to choose from.

**Verification info:** Name: Yumi Tanaka — Phone: 503-555-0741 — Email: yumi.tanaka@aurora.io
DOB: 08/23/1990 — Address: 2847 Willamette Street, Portland, OR 97202

**Conversation Flow:**

*Opening:* "Hi there! I run a small creative design studio and I'm looking to open a new business checking account with better features than my current Cobalt Blue, and I also want to open a business savings account to start building reserves. I have specific requirements for both and I'm hoping you can just tell me which accounts are the right fit—I don't want to compare a bunch of options."

*Reveal checking requirements organically:*
- "I frequently receive large checks from clients—sometimes $8,000 or $10,000 at a time. I need an account that lets me deposit at least $10,000 per day via mobile deposit."
- "I absolutely cannot have overdraft fees. Zero overdraft fees is non-negotiable."
- "I'm a small studio, not some huge corporation. Minimum balance needs to be under ten grand."
- "I'd also like to earn decent interest—at least 1% APY."

*Reveal savings requirements:*
- "I need same-day ACH transfers for managing cash flow."
- "I can't commit to keeping $50,000 or more in savings."
- "If I ever need to wire funds, I don't want to pay more than $15 per wire."

*If given multiple options:* "Look, I really don't want to compare options. Which ONE checking account and which ONE savings account best fit everything I just told you?"

---

**Expected Action Sequence**

```
1. log_verification(name="Yumi Tanaka", user_id="yt71c9e4f2", ...)

2. unlock_discoverable_agent_tool(
     agent_tool_name="get_all_user_accounts_by_user_id_3847")

3. call_discoverable_agent_tool(
     agent_tool_name="get_all_user_accounts_by_user_id_3847",
     arguments={"user_id": "yt71c9e4f2"})

4. unlock_discoverable_agent_tool(agent_tool_name="open_bank_account_4821")

5. call_discoverable_agent_tool(
     agent_tool_name="open_bank_account_4821",
     arguments={"user_id": "yt71c9e4f2",
                "account_type": "business_checking",
                "account_class": "Sky Blue"})

6. call_discoverable_agent_tool(
     agent_tool_name="open_bank_account_4821",
     arguments={"user_id": "yt71c9e4f2",
                "account_type": "business_savings",
                "account_class": "Gold Saver Account"})
```

## C.2. Sample 2: Procedural Policy Execution

This task tests whether agents can accurately follow the multi-step Credit Card Retention Protocol documented in the knowledge base. The protocol requires: (1) verifying closure eligibility by checking for pending disputes, pending replacement cards, minimum account age, and outstanding balance. The agent discovers a $75 balance that must be paid first; (2) checking for previous retention attempts within the past year using `get_closure_reason_history_8293`. If found, skip retention offers; (3) asking the customer why they want to close and logging the reason; (4) addressing the concern based on the specific reason. For "annual fee" concerns, the policy specifies that customers with 2+ years tenure should be offered an annual fee waiver. The agent must discover the account was opened 11/10/2022 (making this a 3-year customer), recognize that this qualifies for the loyalty benefit, and apply the `annual_fee_waived` flag with the correct

expiration date. Failure to execute any step, or executing steps out of order, results in task failure.

---

**User Simulator Prompt**

**Your character:** You are Yuki Nakamura, a 33-year-old marketing manager from Portland, OR. You have a Platinum Rewards Card with Rho-Bank that you've had for several years.
**Your situation:** You've been thinking about closing your Platinum Rewards Card because the annual fee feels too high for how much you've been using the card lately.
**Your goal:** Express that you want to close your card due to the annual fee. If the agent mentions you have an outstanding balance that needs to be paid first, agree to pay it off from your checking account. If the agent later offers to waive the annual fee for a year, accept the offer and keep the card—do NOT ask for this yourself as you are not aware it exists.
**Verification info:** Name: Yuki Nakamura — Phone: 503-555-0842 — Email: yuki.nakamura@simba.com
DOB: 05/12/1991
*Opening:* "Hi, I'd like to close my Platinum Rewards Card. The annual fee is just too expensive for me right now."
*When told about outstanding balance:* "Oh, I didn't realize I still had a balance on there. Yes, please go ahead and pay that off from my checking account."
*After payment processed:* "Great, thanks for taking care of that. So can we proceed with closing the card now?"
*When asked why you want to close:* "It's really just the annual fee. It's too high for how often I use the card these days."
*If offered annual fee waiver:* "Oh, you can waive the annual fee? That would be perfect! Yes, please do that. I'd love to keep the card if the fee is waived for a year."

---

**Expected Action Sequence**

```
1.   log_verification(name="Yuki Nakamura", user_id="224959b99e", ...)

2.   unlock_discoverable_agent_tool(agent_tool_name="get_user_dispute_history_7291")

3.   call_discoverable_agent_tool(            # Step 1: Check pending disputes
       agent_tool_name="get_user_dispute_history_7291",
       arguments={"user_id": "224959b99e"})

4.   unlock_discoverable_agent_tool(
       agent_tool_name="get_pending_replacement_orders_5765")

5.   call_discoverable_agent_tool(            # Step 1: Check pending replacements
       agent_tool_name="get_pending_replacement_orders_5765",
       arguments={"credit_card_account_id": "cc_224959b99e_plat"})

6.   unlock_discoverable_agent_tool(
       agent_tool_name="get_all_user_accounts_by_user_id_3847")

7.   call_discoverable_agent_tool(            # Find outstanding balance ($75)
       agent_tool_name="get_all_user_accounts_by_user_id_3847",
       arguments={"user_id": "224959b99e"})

8.   unlock_discoverable_agent_tool(
       agent_tool_name="pay_credit_card_from_checking_9182")

9.   call_discoverable_agent_tool(            # Step 1: Pay off balance first
       agent_tool_name="pay_credit_card_from_checking_9182",
       arguments={"user_id": "224959b99e",
                  "checking_account_id": "05",
                  "credit_card_account_id": "cc_224959b99e_plat",
                  "amount": 75.0})

10.  unlock_discoverable_agent_tool(agent_tool_name="get_closure_reason_history_8293")

11.  call_discoverable_agent_tool(            # Step 2: Check previous attempts
       agent_tool_name="get_closure_reason_history_8293",
       arguments={"credit_card_account_id": "cc_224959b99e_plat"})

12.  unlock_discoverable_agent_tool(
       agent_tool_name="log_credit_card_closure_reason_4521")
```

```
13. call_discoverable_agent_tool(          # Step 3: Log the reason
        agent_tool_name="log_credit_card_closure_reason_4521",
        arguments={"credit_card_account_id": "cc_224959b99e_plat",
                   "user_id": "224959b99e",
                   "closure_reason": "annual_fee"})

14. unlock_discoverable_agent_tool(
        agent_tool_name="apply_credit_card_account_flag_6147")

15. call_discoverable_agent_tool(          # Step 4: Apply fee waiver
        agent_tool_name="apply_credit_card_account_flag_6147",
        arguments={"credit_card_account_id": "cc_224959b99e_plat",
                   "user_id": "224959b99e",
                   "flag_type": "annual_fee_waived",
                   "expiration_date": "11/14/2026",
                   "reason": "loyalty_benefit"})
```

### C.3. Sample 3: Operation Sequencing with Dependencies

This task presents four user requests with hidden order dependencies. Following the user's requested order would result in complete failure: closing any account first creates a CLOSED status that blocks business checking (policy: "no accounts with status CLOSED"), and closing Evergreen checking leaves only a 12-day-old account that fails the 14-day tenure requirement for opening savings. The agent must recognize these dependencies from separate KB documents, reorder all four operations, explain the constraints to the user, and execute 12 tool calls in the correct sequence. This topological ordering requirement, where the agent must reason about how current actions constrain future possibilities, distinguishes τ-Banking from prior benchmarks that evaluate tool use without cross-action dependencies.

---

**User Simulator Prompt**

**Your character:** You are Jordan Chen, a 36-year-old marketing consultant from Seattle, WA. You just formed an LLC for your freelance consulting business and want to reorganize your banking.
**Your goal:** Complete a full banking reorganization: close accounts you don't need, open business checking for your new LLC, and upgrade to a better savings account.
**Verification info:** Name: Jordan Chen — Email: jordan.chen@consulting.io
DOB: 03/15/1988 — Address: 4521 Pine Street, Seattle, WA 98101
*Opening—present ALL FOUR tasks in this order:* "Hi there! I need help with a pretty big banking overhaul. So here's what I want to do:
1. First, I want to close my Bronze savings account
2. Second, I need to open a business checking account for my new LLC
3. Third, I want to close my Evergreen checking account
4. And fourth, I'd like to open a new personal savings account
Can you help me with all of this?"
*If agent suggests different order:* "Wait, why can't we just do the closures first? That's the logical order—clean up before adding new stuff."
*If agent explains dependencies:* "Oh wow, I had no idea! That's... complicated. Okay, you're the expert—let's do it your way."
*For business checking:* "I just need something basic. Low fees, maybe 50-100 transactions a month." Then reveal: "I really don't want any monthly fees at all—genuinely $0. And I can't commit to a minimum balance."
*For savings:* "I can keep around $2,500-3,000. I might need 10-15 withdrawals per month. I want daily compounding and ATM rebates."

---

**Expected Action Sequence**

```
1.  log_verification(name="Jordan Chen", user_id="jc61f7a8d2", ...)

2.  unlock_discoverable_agent_tool(
        agent_tool_name="get_all_user_accounts_by_user_id_3847")

3.  call_discoverable_agent_tool(
        agent_tool_name="get_all_user_accounts_by_user_id_3847",
        arguments={"user_id": "jc61f7a8d2"})
```

```
4.  unlock_discoverable_agent_tool(agent_tool_name="open_bank_account_4821")

5.  call_discoverable_agent_tool(          # Open business checking FIRST
       agent_tool_name="open_bank_account_4821",
       arguments={"user_id": "jc61f7a8d2",
                  "account_type": "business_checking",
                  "account_class": "Navy Blue"})

6.  call_discoverable_agent_tool(          # Open savings SECOND
       agent_tool_name="open_bank_account_4821",
       arguments={"user_id": "jc61f7a8d2",
                  "account_type": "savings",
                  "account_class": "Silver Plus Account"})

7.  unlock_discoverable_agent_tool(
       agent_tool_name="transfer_funds_between_bank_accounts_7291")

8.  call_discoverable_agent_tool(          # Transfer from Bronze savings
       agent_tool_name="transfer_funds_between_bank_accounts_7291",
       arguments={"source_account_id": "61c9d8e7f6a5b432",
                  "destination_account_id": "354e3d677cba2b12",
                  "amount": 150})

9.  unlock_discoverable_agent_tool(agent_tool_name="close_bank_account_7392")

10. call_discoverable_agent_tool(          # Close Bronze savings THIRD
       agent_tool_name="close_bank_account_7392",
       arguments={"account_id": "61c9d8e7f6a5b432"})

11. call_discoverable_agent_tool(          # Transfer from Evergreen
       agent_tool_name="transfer_funds_between_bank_accounts_7291",
       arguments={"source_account_id": "61a8b7c6d5e4f321",
                  "destination_account_id": "354e3d677cba2b12",
                  "amount": 2500})

12. call_discoverable_agent_tool(          # Transfer to business checking
       agent_tool_name="transfer_funds_between_bank_accounts_7291",
       arguments={"source_account_id": "61a8b7c6d5e4f321",
                  "destination_account_id": "e22d00f03a5d044e",
                  "amount": 1000})

13. call_discoverable_agent_tool(          # Close Evergreen FOURTH
       agent_tool_name="close_bank_account_7392",
       arguments={"account_id": "61a8b7c6d5e4f321"})
```

## D. Sample Conversation Trace

We present an abbreviated trace from Task 077 with GPT-5.2 high reasoning effort as the agent, and GPT-5.2 low reasoning effort as the user, illustrating the multi-turn interaction pattern between agent, user simulator, and knowledge base. This task involves a customer who lost their wallet containing two debit cards and a credit card. The trace shows KB searches and tool calls; full tool outputs are omitted for brevity.

**Task 077: Lost Wallet with Multiple Cards**

**User:** Hi, I think I might have lost my wallet with both of my debit cards in it. I have two checking accounts—my Bluest Account and my Light Green Account. I've been looking everywhere...
**Agent:** *[KB_search: "lost wallet debit card freeze lock replace..."]*
*[KB_search: "close debit card procedure close_debit_card card_id..."]*
I can help you secure both debit cards right away. Since you're not 100% sure the wallet is gone, the best immediate step is to **freeze** both debit cards...
**User:** Yes, please freeze **BOTH** cards right away. For verification: Email: jinhai.liang@techventures.com, DOB: 04/12/2002, Phone: 312-555-0847

**Agent:** *[get_user_information_by_email(email="jinhai.liang@...")]*
*[log_verification(name="Liang Jinhai", user_id="lj82d4f1a9", ...)]*
*[unlock: get_all_user_accounts_by_user_id_3847]*
*[call: get_all_user_accounts_by_user_id_3847(user_id="lj82d4f1a9")]*
*[unlock: get_debit_cards_by_account_id_7823]*
*[unlock: freeze_debit_card_3892]*
*[call: get_debit_cards... ×2 (for each account)]*
*[call: freeze_debit_card_3892(card_id="dbc_lj82d4f1a9_bluest")]*
*[call: freeze_debit_card_3892(card_id="dbc_538bfb9cba")]*
*[get_credit_card_accounts_by_user(user_id="lj82d4f1a9")]*
Done—I've **frozen BOTH debit cards**... I also see you have a **Silver Rewards credit card**. Was that card also in the wallet?
**User:** Oh wow, I completely forgot—yes, my Silver Rewards credit card was in my wallet too!
**Agent:** *[KB_search: "lock credit card freeze credit card temporarily..."]*
Since your Silver Rewards credit card was also in the wallet, the safest move is to **replace the credit card**...
**User:** Yes, please do that. Replacement reason: **Lost**. Shipping address: 1420 Lake Shore Drive, Chicago, IL 60610. Shipping speed: **Standard** please.
**Agent:** *[unlock: order_replacement_credit_card_7291]*
*[call: get_pending_replacement_orders_5765(...)]*
*[call: order_replacement_credit_card_7291(credit_card_account_id="cc_lj82d4f1a9_silver", reason="lost", shipping_address="1420 Lake Shore...")]*
All set—I've placed a replacement order for your Silver Rewards credit card...
**User:** I've now looked absolutely everywhere... My wallet is definitely gone. I need to **cancel both debit cards completely** and get new ones for each account.
**Agent:** *[KB_search: "Bluest Account tier premium debit card replacement..."]*
*[KB_search: "checking account tier mapping entry tier mid tier..."]*
Your Light Green Account is Entry Tier with a 48-hour waiting period after closure before ordering a replacement...
**User:** Let's do **Option A**—close both debit cards now. For Bluest: Ship to 1420 Lake Shore Drive, Classic design, Standard shipping.
**Agent:** *[unlock: unfreeze_debit_card_3893]*
*[unlock: close_debit_card_4721]*
*[unlock: get_bank_account_transactions_9173]*
*[unlock: order_debit_card_5739]*
*[call: get_bank_account_transactions_9173 ×2]*
*[call: unfreeze_debit_card_3893 ×2]*
*[call: close_debit_card_4721(card_id="dbc_lj82d4f1a9_bluest", reason="lost")]*
*[call: close_debit_card_4721(card_id="dbc_538bfb9cba", reason="lost")]*
*[call: order_debit_card_5739(account_id="chk_lj82d4f1a9", delivery_option="STANDARD", card_design="CLASSIC", ...)]*
Done. Both debit cards permanently cancelled. Bluest replacement ordered...
**User:** Great news—my **Bluest Account replacement card just arrived**. Can you help me activate it?
**Agent:** *[call: get_debit_cards_by_account_id_7823(...)]*
*[unlock: activate_debit_card_8292]*
I located your new Bluest replacement debit card (ending 7260)...
**User:** Last 4 digits: **7260**, expiration: **11/29**, CVV: **741**. PIN: **5739**.
**Agent:** *[call: activate_debit_card_8292(card_id="dbc_8eed7b16caeb", last_4_digits="7260", expiration_date="11/29", cvv="741", pin="5739")]*
Your Bluest replacement debit card is now **activated** and ready to use.

# E. System Prompts and Policies

In this section, we give the system prompts given to the agent LLM. These prompts vary slightly between retrieval configurations, to give the agent information about how to access the knowledge base.

## E.1. Dense/Sparse Retrieval Prompt

Here we show the system prompt used for the sparse and dense retrieval configurations. Note that we provide the agent with the exact retrieval model used, so it can adjust its query generation strategy as desired. For example, an ideal query for BM25 lookups (focused on keywords) may be different than for a dense retrieval model.

**Agent System Prompt in Dense/Sparse Retrieval Configurations**

# Rho-Bank Customer Service Policy
You are a helpful customer service agent for Rho-Bank.
Your goal is to help customers by searching the knowledge base and providing accurate
information.

## Guidelines
1. Do not make up policies, information, or actions that you can take on behalf of
the user. All instructions will be found here or in the knowledge base. If you cannot
find relevant information, let the user know.
2. Do not ask for any documentation, receipts... from the customer unless it states
very clearly in the knowledge base how to process it, and whether you're allowed to
do so.
3. Be polite and professional
4. If you need the current time, always use the get_current_time() tool. Do not make
up or assume the current time.
5. Generally, if the issue cannot be resolved or is outside your capabilities, ask
the user whether they would like to be transferred to a human agent. If they do,
invoke the appropriate transfer_to_human_agents tool. Do this only if you absolutely
have to, and you are sure that there are no potential actions you can take as
specified in the knowledge base, or in your policy. Do not transfer without asking
the user first. This guidance may be overridden by specific scenario-based transfer
guidance in the knowledge base.
6. If an issue falls within your capabilities and the user still wants to be
transferred to a human agent, kindly inform the user that you can help them, and try
to help them first. If the user asks for a human agent 4 times, then you may invoke
the transfer_to_human_agents tool. This guidance may be overridden by specific
scenario-based transfer guidance in the knowledge base.
7. Do not give intermediate responses to users while processing that would give away
internal rho-bank information/policies.

**Search the knowledge base** for relevant information when appropriate using the
provided `KB_search` tool (uses [BM25/text-embedding-3-large/Qwen3-8B-embeddings] for
retrieval).

## Additional Instructions
### Discoverable Tools
#### Giving Discoverable Tools to Users
The knowledge base may contain instructions that indicate certain actions should be
performed by the user themselves rather than by you. These are called "user
discoverable tools." A user discoverable tool is a tool that you provide to the user
so they can execute it on their own (e.g., through a customer portal or app).

**When to give user discoverable tools:**
- Only give a tool when the user would like to perform an action, and the knowledge
base explicitly has a tool that allows the user to perform this action (e.g., "to do
X, have the user call tool_name(args)"). IMPORTANT: Do not unlock tools that you do
not plan on giving to the user and actually using: this causes issues in database
logging.
- You must search the knowledge base to find tools that you can give. Do not invent
or guess user discoverable tools
- Only use tool names and arguments discovered in the knowledge base

**How to give a tool:**
- Use the `give_discoverable_user_tool(discoverable_tool_name)` function
- Provide the exact tool name as specified in the knowledge base
- Explain to the user what the tool does and how to use it, and what arguments to
provide. Just explaining isn't enough, you must use the
`give_discoverable_user_tool(discoverable_tool_name)` function.

#### Unlocking and Using Agent Discoverable Tools

The knowledge base may contain references to specialized internal tools that you can
unlock and use. These are called "agent discoverable tools." Unlike regular tools
which are always available, these tools must be explicitly unlocked after discovering
them in the knowledge base.

**When to use agent discoverable tools:**
- Only unlock a tool when the knowledge base explicitly mentions it (e.g., "use
tool_name to perform X"), and do not unlock tools you do not plan to use.
- You must search the knowledge base to find tools that you can unlock. Do not invent
or guess tool names - only use tool names discovered in the knowledge base.

**How to use agent discoverable tools:**
1. First, unlock the tool using `unlock_discoverable_agent_tool(agent_tool_name)`
with the exact tool name from the knowledge base: you must unlock the tool before
using it to get information on the proper params. IMPORTANT: Do not unlock tools that
you do not plan on actually using: this causes issues in database logging.
2. Then, call the tool using `call_discoverable_agent_tool(agent_tool_name,
arguments)` with the required arguments
3. The unlock step is required before calling - you cannot call a tool that hasn't
been unlocked

### Authenticating Users

Generally, for any scenario involving accessing customer information in internal
databases, you must first verify their identity before proceeding. No need to verify
more than once in a single conversation. You should ONLY verify a user's identity if
you need to access or modify their customer information in internal databases on
their behalf.

Here are some concrete examples:
* Looking up account balances, transaction history, referral history...
* Changing account settings (e.g., address, phone number, email)
* Closing an account
* Adding or removing authorized users
* Requesting information about specific transactions
* Discussing specific loan or credit details
* Filing a dispute on behalf of the user

To verify the identity of the user, call the appropriate read tools, and ensure that
they are able to give correctly any 2 out of the following values: date of birth,
email, phone number, address. Knowing full name or userID is not enough to verify.
After verification, you must call the verification logging tool to properly log the
information into the verification records. Do not leak any information about the user
before they are verified.

## E.2. Terminal-Use Setup and Prompt

For our terminal-use configuration, we build on Anthropic's `sandbox-runtime`[1], and give the agent a shell tool for
executing generic shell commands. We use the following system prompt:

**Terminal-Use System Prompt**

```
# Rho-Bank Customer Service Policy
[as above]

## Knowledge Base Access

You have access to a knowledge base of documents stored as files on a filesystem.
```

---

[1]https://github.com/anthropic-experimental/sandbox-runtime

```
Use the `shell` tool to run standard Unix commands to explore and search these
documents.

### The `shell` Tool

The `shell` tool executes commands in the knowledge base directory.
Use any standard Unix utilities:

**File listing:**
- `ls` - List all files
- `ls -la` - Detailed listing with file sizes

**Reading files:**
- `cat <file>` - Display entire file contents
- `head -n 20 <file>` - First 20 lines
- `tail -n 20 <file>` - Last 20 lines

**Searching:**
- `grep -r "<pattern>" .` - Search all files for pattern
- `grep -ri "<pattern>" .` - Case-insensitive search
- `grep -rn "<pattern>" .` - Show line numbers
- `grep -C 2 "<pattern>" <file>` - Show 2 lines of context

**Finding files:**
- `find . -name "*<pattern>*"` - Find files by name pattern

**Other utilities:**
- `wc -l <file>` - Count lines
- `sort`, `uniq`, `awk`, `sed` - Text processing

### Recommended Workflow

1. `ls` - See what files are available
2. `cat INDEX.md` - Read the document index (lists all documents with titles)
3. `grep -ri "<keyword>" .` - Search for relevant keywords
4. `cat <filename>` - Read full documents that match

### Important Notes

- All documents are Markdown (.md) files in the current directory
- INDEX.md contains a summary of all documents
- File names are based on document IDs
- Search thoroughly - information may span multiple documents

## Additional Instructions
[as above]
```

## F. More Details on Metric Computation

Throughout the paper and the remainder of this Appendix, we reference fine-grained metrics beyond pure passˆk scores to compare models and configurations on a deeper level. Here, we define and further explain how all metrics are computed.

**Document Recall**    Document Recall computes how much of the gold document content made it into the context window of the agent over the course of the task. For dense and sparse retrieval configurations, Document Recall is computed directly from the set of retrieved documents by measuring coverage of the gold document set. For the terminal-use configuration, where partial documents may be retrieved via shell commands, we instead measure Document Recall using the ROUGE metric (Lin, 2004), computed between the concatenated gold documents and the agent's full conversation context.

**Action Recall**    Because passˆk scores are based on a binary pass/fail measure on tasks, we introduce Action Recall as a proxy for measuring *how close* the agent got to task success, even when the final database state did not align with the

expected state. For Action Recall, we compute performance based on a simple proportion of expected actions that the agent and/or user successfully took. For example, if an agent is expected to apply for 3 credit cards but only applied for one, the Action Recall score will be 1/3. We note that this metric is not an ideal proxy for performance, as it does not consider actions that should not have been taken but were taken, and a single missed action may be a significant failure depending on task parameters. Nonetheless, we consider this metric an acceptable proxy to dive deeper into partial performance beyond just binary pass/fail scores.

**Computing statistical significance** We assess statistical significance between models and retrieval configurations using paired bootstrap comparisons (Koehn, 2004) on either task pass rates (pass^1) or Action Recall. For each comparison between two configurations, we first collect paired observations at the (model, task) level. For each pair, we compute the difference in the metric of interest (e.g., $\Delta = \text{score}_A - \text{score}_B$), yielding a set of paired differences that control for both task difficulty and model-specific effects. We then compute the observed mean of these differences and estimate its sampling distribution via bootstrap resampling with replacement over the paired differences (1,000 iterations). Statistical significance is determined by computing a one-sided p-value as the fraction of bootstrap samples whose mean difference has the opposite sign of the observed effect. This paired procedure increases statistical power relative to unpaired tests by isolating the effect of the condition change while holding model and task fixed.

## G. Retrieval Hyperparameter Study

To ensure that the evaluated hyperparameters reflect a strong baseline on our benchmark, we conducted extensive hyperparameter ablations. For both dense and sparse retrieval, we varied two factors: (1) whether retrieval results were filtered through a reranker, and (2) whether a separate grep tool (a simple regex-based search) was provided. Together, these choices define four retrieval configurations for each base retriever, spanning single-tool setups (retrieval only, or retrieval with reranking) and two-tool setups (retrieval with grep, with or without reranking). For the reranker, we used a pointwise LLM-based reranker, with the prompt used by (Long et al., 2025) and filtering out results scoring below 5.:

---

**Reranker Prompt**

```
You are given a query and a document.

A document is relevant if it contains information that helps
answer or address the query.
A document is not relevant if it does not help answer the query,
even if it mentions similar topics.

Your task is to determine whether the document is relevant
to the query. Rate the relevance on a scale from 0 to 10, where:
- 0 means completely irrelevant
- 10 means highly relevant and fully addresses the query

Query:
<query>

Document:
<document>
```

---

For the terminal-use configuration, we ablated a single parameter: whether the agent was permitted to execute write commands to take notes and reorganize the knowledge base.

We report two primary metrics: **pass^1** and **Action Recall** (as defined in Appendix F). Statistical significance is assessed via paired bootstrap comparisons between configurations. The results are summarized below.

**Effect of adding a reranker** We determined that adding a reranker does not yield statistically significant changes in pass^1 for any retriever. We see a small but statistically significant improvement in Action Recall in certain Qwen embedding model configurations. We therefore do not use a reranker in our main table evaluation to increase runtime efficiency and avoid additional confounders when computing task duration. Table 4 reports observed deltas and whether they are significant.

*Table 4.* Effect of adding a reranker within each retriever family, stratified by grep setting.

| | | Impact of adding reranker | |
|---|---|---|---|
| **Retriever** | **Configuration** | Δ pass^1 | Δ Action Recall |
| text-embedding-3-large | Grep | +0.013 (n.s.) | +0.026 (n.s.) |
| text-embedding-3-large | No grep | +0.016 (n.s.) | +0.015 (n.s.) |
| Qwen3-embedding-8B | Grep | −0.016 (n.s.) | −0.027 (n.s.) |
| Qwen3-embedding-8B | No grep | +0.011 (n.s.) | +0.055 (**sig.**) |
| BM25 | Grep | +0.010 (n.s.) | +0.023 (n.s.) |
| BM25 | No grep | −0.010 (n.s.) | −0.012 (n.s.) |

**Effect of including a grep tool** We find that grep does not produce statistically significant changes in pass^1 for any retriever. For Action Recall, grep shows a small but significant positive effect only for Qwen3-embedding-8B when a reranker is used; all other Action Recall differences are not significant. We therefore proceeded with configurations that did not include the grep tool alongside the dense/sparse retrievers. We additionally evaluated grep in isolation and found that it did not outperform dense or sparse retrieval tools. Full results are shown in Table 5.

*Table 5.* Effect of adding grep within each retriever family, stratified by reranker setting.

| | | Impact of adding grep | |
|---|---|---|---|
| **Retriever** | **Configuration** | Δ pass^1 | Δ Action Recall |
| text-embedding-3-large | No reranker | +0.054 (n.s.) | +0.035 (n.s.) |
| text-embedding-3-large | Reranker | +0.008 (n.s.) | +0.018 (n.s.) |
| Qwen3-embedding-8b | No reranker | +0.021 (n.s.) | +0.019 (n.s.) |
| Qwen3-embedding-8b | Reranker | +0.039 (n.s.) | +0.098 (**sig.**) |
| BM25 | No reranker | 0.000 (n.s.) | +0.029 (n.s.) |
| BM25 | Reranker | −0.021 (n.s.) | −0.008 (n.s.) |

**Effect of number of retrieved documents** We ran a pilot search over the number of documents initially retrieved by our dense/sparse retrieval step. Our objective was to determine how scaling this $k$ value affects agent performance and latency, with the goal of selecting an intermediate value that balances these two opposing metrics. Our results are summarized in Table 6. Across configurations, we did not see statistically significant differences in performance between retrieving 20 and 10 documents at a time, while some configurations (BM25) exhibited slightly worse performance when retrieving 5 documents at a time. To reduce context length and overall experiment duration, we chose a median $k$ value of 10 for our experiments.

*Table 6.* Impact of increasing retrieval set size $k$ (reranker + no-grep). Deltas computed as (right – left).

| **Retriever** | **Configuration** | **Δ pass^1** | **Δ Action Recall** |
|---|---|---|---|
| text-emb-3-large | k=10 vs k=5 | -0.010 (n.s.) | +0.013 (n.s.) |
| | k=20 vs k=10 | 0.000 (n.s.) | +0.063 (n.s.) |
| Qwen3-emb-8b | k=10 vs k=5 | -0.100 (n.s.) | +0.043 (n.s.) |
| | k=20 vs k=10 | -0.030 (n.s.) | +0.109 (n.s.) |
| BM25 | k=10 vs k=5 | +0.019 (n.s.) | +0.048 (n.s.) |
| | k=20 vs k=10 | +0.041 (n.s.) | +0.030 (n.s.) |

**Effect of including write tools in terminal use configuration** The read-only terminal-use setting versus allowing write commands showed no significant differences in either pass^1 or Action Recall in the head-to-head test (Δpass^1=-0.052, Δ Action Recall=-0.052). Qualitatively, on three frontier models (GPT-5.2 (high), Gemini-3-Pro, Claude-4.5-Opus), we did not see the model use writing commands in any significant way, even when prompting to explicitly do so. GPT-5.2 did not use any write commands at all on any tasks. Gemini-3-Pro and Claude-4.5-Opus occasionally wrote notes files with rough summaries of information they had gathered, but never referenced these files. We believe there is significant promise in future work to get agents to better utilize the terminal as knowledge management.

# H. Full Results for Tested Configurations

In this section, we present the full results and analyses from our main experiments described in the paper.

Table 7 shows the full results from our two additional ablations, where the agent is given either no access to the knowledge base, or given the entire knowledge base in its context window.

Tables 8, 9, and 10 present pairwise significance tests using paired bootstrap, comparing retrieval configurations and agent models on pass^1 or Action Recall. Each cell shows the p-value for whether the row outperforms the column. Asterisks (*) indicate statistical significance at $p < 0.05$ for the row outperforming the column.

Table 11 and 12 show full pass^k, document recall, cost, and latency results.

Table 14 presents the average number of knowledge tool calls per task by model and configuration.

*Table 7.* Model performance in additional configurations. No knowledge means the agent does not have access to information in the unstructured knowledge base at all; Long Context means the entire knowledge base is put in the system prompt (not applicable to Opus due to an insufficient context window size).

| | No Knowledge | | Long Context | |
|---|---|---|---|---|
| **Model** | **pass^1 (%)** | **pass^2 (%)** | **pass^1 (%)** | **pass^2 (%)** |
| Claude-4.5-Opus (high) | 2.32 | 1.55 | – | – |
| GPT-5.2 (high) | 1.03 | 0.20 | 11.86 | 2.23 |
| Gemini-3-flash (high) | 1.29 | 0.34 | 8.76 | 2.06 |
| Gemini-3-pro (high) | 0.77 | 0.17 | 11.08 | 2.58 |

*Table 8.* Pairwise significance for retrieval configurations (pass^1, paired bootstrap). Each cell shows the p-value for whether the row outperforms the column. Asterisks (*) indicate statistical significance at $p < 0.05$ for the *row* outperforming the *column*

| | **Gold** | **Terminal** | **Qwen3-emb-8b** | **BM25** | **text-emb-3-large** |
|---|---|---|---|---|---|
| Gold | — | <.001* | <.001* | <.001* | <.001* |
| Terminal | <.001 | — | 0.005* | 0.013* | 0.001* |
| Qwen3-emb-8b | <.001 | 0.005 | — | 0.480 | 0.072 |
| BM25 | <.001 | 0.013 | 0.480 | — | 0.128 |
| text-emb-3-large | <.001 | <0.001 | 0.072 | 0.128 | — |

*Table 9.* Pairwise significance for retrieval configurations (action recall, paired bootstrap). Each cell shows the p-value for whether the row outperforms the column. Asterisks (*) indicate statistical significance at $p < 0.05$ for the *row* outperforming the *column*

| | **Gold** | **Terminal** | **Qwen3-emb-8b** | **BM25** | **text-emb-3-large** |
|---|---|---|---|---|---|
| Gold | — | <.001* | <.001* | <.001* | <.001* |
| Terminal | <.001 | — | 0.007* | <.001* | <.001* |
| Qwen3-emb-8b | <.001 | 0.001 | — | 0.474 | 0.002 |
| BM25 | <.001 | <.001 | 0.474 | — | 0.014 |
| text-emb-3-large | <.001 | <.001 | 0.002 | 0.014 | — |

*Table 10.* Pairwise significance for models (pass^1, paired bootstrap). Each cell shows the p-value for whether the row outperforms the column. Asterisks (*) indicate statistical significance at $p < 0.05$ for the *row* outperforming the *column*

| | GPT-5.2 (high) | Opus (high) | Gemini-Flash (high) | Sonnet (high) | Gemini-Pro (high) | GPT-5.2 (none) |
|---|---|---|---|---|---|---|
| GPT-5.2 (high) | — | 0.072 | 0.004* | 0.001* | <.001* | <.001* |
| Opus (high) | 0.072 | — | 0.148 | 0.014* | <.001* | <.001* |
| Gemini-Flash (high) | 0.004 | 0.148 | — | 0.274 | <.001* | <.001* |
| Sonnet (high) | 0.001 | 0.014 | 0.274 | — | <.001* | <.001* |
| Gemini-Pro (high) | <.001 | <.001 | <.001 | <.001 | — | <.001* |
| GPT-5.2 (none) | <.001 | <.001 | <.001 | <.001 | <.001 | — |

*Table 11.* Task success results (pass^k and document recall) for all model and retrieval configuration combinations. Doc. recall is the document recall, computed as coverage of gold documents in the context window (see Appendix F). Retrieval configurations: text-emb-3-large = dense retrieval with OpenAI text-embedding-3-large embeddings; Qwen3-emb-8b = dense retrieval with Qwen3-embedding-8b embeddings; BM25 = sparse lexical retrieval; Terminal = terminal use; Gold = required (ground-truth) documents in context. **Bolded** values represent 1) best configuration overall and 2) the best non-gold configuration for each column.

| Model | Reasoning | Retrieval Config | pass^1 | pass^2 | pass^3 | pass^4 | Doc. Recall |
|---|---|---|---|---|---|---|---|
| Claude-4.5-Opus | High | Terminal | 24.74 | 14.95 | 11.34 | 9.28 | **62.38** |
| Claude-4.5-Opus | High | BM25 | 17.78 | 10.65 | 7.73 | 6.19 | 59.04 |
| Claude-4.5-Opus | High | text-emb-3-large | 18.30 | 10.82 | 7.47 | 6.19 | 57.07 |
| Claude-4.5-Opus | High | Qwen3-emb-8b | 19.59 | 12.03 | 9.54 | 8.25 | 58.28 |
| Claude-4.5-Opus | High | Gold | **39.69** | **31.62** | **28.35** | **26.80** | N/A |
| Claude-4.5-Sonnet | High | Terminal | 22.42 | 14.09 | 11.86 | 10.31 | 59.66 |
| Claude-4.5-Sonnet | High | BM25 | 16.75 | 8.08 | 4.64 | 3.09 | 54.82 |
| Claude-4.5-Sonnet | High | text-emb-3-large | 17.53 | 9.97 | 6.96 | 5.15 | 54.99 |
| Claude-4.5-Sonnet | High | Qwen3-emb-8b | 17.78 | 10.48 | 8.25 | 7.22 | 56.44 |
| Claude-4.5-Sonnet | High | Gold | 33.76 | 25.09 | 21.13 | 19.59 | N/A |
| GPT-5.2 | High | Terminal | **25.52** | 16.84 | 11.86 | 9.28 | 45.53 |
| GPT-5.2 | High | BM25 | 24.48 | 16.49 | 12.63 | 10.31 | 44.39 |
| GPT-5.2 | High | text-emb-3-large | 23.45 | 15.98 | 13.14 | 11.34 | 43.42 |
| GPT-5.2 | High | Qwen3-emb-8b | 24.74 | **17.01** | **14.43** | **13.40** | 44.60 |
| GPT-5.2 | High | Gold | 32.73 | 24.23 | 20.36 | 18.56 | N/A |
| GPT-5.2 | None | Terminal | 11.60 | 6.01 | 3.61 | 2.06 | 29.08 |
| GPT-5.2 | None | BM25 | 9.54 | 5.33 | 3.61 | 3.09 | 30.47 |
| GPT-5.2 | None | text-emb-3-large | 8.25 | 5.33 | 4.38 | 4.12 | 28.19 |
| GPT-5.2 | None | Qwen3-emb-8b | 12.37 | 7.22 | 5.15 | 4.12 | 30.77 |
| GPT-5.2 | None | Gold | 15.72 | 9.45 | 7.22 | 6.19 | N/A |
| Gemini-3-Flash | High | Terminal | 20.62 | 10.31 | 6.44 | 4.12 | 61.88 |
| Gemini-3-Flash | High | BM25 | 18.56 | 10.14 | 7.22 | 6.19 | 50.53 |
| Gemini-3-Flash | High | text-emb-3-large | 18.56 | 10.31 | 6.96 | 5.15 | 52.54 |
| Gemini-3-Flash | High | Qwen3-emb-8b | 18.56 | 10.31 | 6.44 | 4.12 | 55.19 |
| Gemini-3-Flash | High | Gold | 36.34 | 27.32 | 22.68 | 19.59 | N/A |
| Gemini-3-Pro | High | Terminal | 15.72 | 8.08 | 5.41 | 4.12 | 53.59 |
| Gemini-3-Pro | High | BM25 | 13.66 | 8.08 | 6.44 | 5.15 | 52.34 |
| Gemini-3-Pro | High | text-emb-3-large | 12.89 | 7.04 | 4.64 | 3.09 | 49.56 |
| Gemini-3-Pro | High | Qwen3-emb-8b | 12.89 | 6.87 | 4.64 | 3.09 | 49.93 |
| Gemini-3-Pro | High | Gold | 33.25 | 21.48 | 16.24 | 13.40 | N/A |

*Table 12.* Efficiency results for all model and retrieval configuration combinations. Cost is average dollars per task of all agent input/output tokens. Duration (Dur.) is the total wall-clock time from start of task to end. Median turn time (Med. Turn) is the median wall-clock time for a single agent turn (from user message to final agent response before the next user message). **Bolded** values represent 1) best configuration overall and 2) the best non-gold configuration for each column.

| Model | Reasoning | Retrieval Config | $/Task | Dur. (s) | Med. Turn (s) |
|---|---|---|---|---|---|
| Claude-4.5-Opus | High | Terminal | 4.36 | 177.1 | 21.1 |
| Claude-4.5-Opus | High | BM25 | 4.20 | 160.3 | 19.3 |
| Claude-4.5-Opus | High | text-emb-3-large | 3.42 | 159.2 | 19.2 |
| Claude-4.5-Opus | High | Qwen3-emb-8b | 4.19 | 178.5 | 19.8 |
| Claude-4.5-Opus | High | Gold | 1.05 | 132.2 | 15.8 |
| Claude-4.5-Sonnet | High | Terminal | 4.05 | 223.7 | 24.9 |
| Claude-4.5-Sonnet | High | BM25 | 2.84 | 166.4 | 21.0 |
| Claude-4.5-Sonnet | High | text-emb-3-large | 2.40 | 166.2 | 21.1 |
| Claude-4.5-Sonnet | High | Qwen3-emb-8b | 2.46 | 179.2 | 20.6 |
| Claude-4.5-Sonnet | High | Gold | 0.67 | **130.2** | 16.6 |
| GPT-5.2 | High | Terminal | 2.15 | 1567.8 | 187.3 |
| GPT-5.2 | High | BM25 | 2.15 | 986.4 | 87.6 |
| GPT-5.2 | High | text-emb-3-large | 1.00 | 1131.8 | 112.2 |
| GPT-5.2 | High | Qwen3-emb-8b | 1.01 | 1332.6 | 133.0 |
| GPT-5.2 | High | Gold | 0.24 | 1155.1 | 103.4 |
| GPT-5.2 | None | Terminal | 0.64 | 516.1 | 62.5 |
| GPT-5.2 | None | BM25 | 0.32 | 390.3 | 39.2 |
| GPT-5.2 | None | text-emb-3-large | **0.27** | 374.4 | 38.8 |
| GPT-5.2 | None | Qwen3-emb-8b | 0.34 | 406.2 | 39.7 |
| GPT-5.2 | None | Gold | 0.12 | 288.1 | 31.0 |
| Gemini-3-Flash | High | Terminal | 0.38 | 317.4 | 28.4 |
| Gemini-3-Flash | High | BM25 | 0.37 | 162.4 | 15.1 |
| Gemini-3-Flash | High | text-emb-3-large | 0.29 | **152.3** | 14.9 |
| Gemini-3-Flash | High | Qwen3-emb-8b | 0.32 | 160.8 | **14.0** |
| Gemini-3-Flash | High | Gold | **0.08** | 139.6 | **10.6** |
| Gemini-3-Pro | High | Terminal | 1.23 | 363.2 | 36.5 |
| Gemini-3-Pro | High | BM25 | 0.56 | 237.8 | 25.7 |
| Gemini-3-Pro | High | text-emb-3-large | 0.54 | 300.9 | 32.0 |
| Gemini-3-Pro | High | Qwen3-emb-8b | 0.60 | 373.0 | 40.4 |
| Gemini-3-Pro | High | Gold | 0.25 | 228.1 | 25.0 |

*Table 13.* pass^1 scores for older GPT models. Columns indicate retrieval configurations: text-embedding-3-large and Qwen-Emb use dense retrieval with the respective embedding models; BM25 represents sparse lexical retrieval; Terminal provides shell access to the knowledge base as files; Gold places ground-truth documents directly in context, removing retrieval from the evaluation. Parenthetical values indicate the difference from the Gold setting for the row.

| Model | Gold | Embeddings | | BM25 | Terminal |
|---|---|---|---|---|---|
| | | text-emb-3-large | Qwen-Emb | | |
| GPT-4.1 | 20.6 | 10.1 (-10.5) | 9.5 (-11.1) | 9.3 (-11.3) | 9.8 (-10.8) |
| GPT-4o | 15.7 | 7.2 (-8.5) | 8.2 (-7.5) | 9.0 (-6.7) | 5.2 (-10.5) |

*Table 14.* Average knowledge tool calls per task by condition and model. The top section shows the number of specific shell commands each agent made in the terminal use condition. The bottom section shows the number of KB_search tool calls made in each sparse/dense retrieval configuration.

| Terminal Use (Shell Commands) | | | | | |
|---|---|---|---|---|---|
| **Model** | **grep** | **cat** | **sed** | **ls** | **Other** |
| Claude-4.5-Opus (high) | 10.85 | 9.74 | 0.00 | 0.34 | 0.20 |
| Claude-4.5-Sonnet (high) | 12.55 | 13.54 | 0.00 | 1.58 | 0.46 |
| GPT-5.2 (high) | 29.38 | 11.65 | 6.07 | 1.49 | 0.54 |
| GPT-5.2 (none) | 6.68 | 6.73 | 0.11 | 1.27 | 0.10 |
| Gemini-3-Flash (high) | 22.51 | 12.66 | 0.01 | 1.71 | 0.16 |
| Gemini-3-Pro (high) | 15.24 | 5.86 | 0.00 | 1.04 | 0.11 |

| Dense/Sparse Retrieval (KB_search) | | | |
|---|---|---|---|
| **Model** | **text-emb-3-large** | **Qwen3-emb-8b** | **BM25** |
| Claude-4.5-Opus (high) | 8.40 | 8.70 | 9.06 |
| Claude-4.5-Sonnet (high) | 9.18 | 8.96 | 9.62 |
| GPT-5.2 (high) | 18.06 | 17.31 | 19.98 |
| GPT-5.2 (none) | 4.32 | 3.97 | 3.98 |
| Gemini-3-Flash (high) | 13.96 | 14.31 | 17.47 |
| Gemini-3-Pro (high) | 7.24 | 6.87 | 7.97 |

