# OpenReview forum: "$\tau$-Knowledge: Evaluating Conversational Agents over Unstructured Knowledge"
_ICML.cc/2026/Conference — ICML 2026 regular_

### Official Review · Reviewer_8hyr · 2026-02-15

**Soundness:** 2
**Presentation:** 2
**Significance:** 2
**Originality:** 2
**Overall Recommendation:** 3
**Confidence:** 4

**Summary:**

The paper introduces a new benchmark $\tau$-Knowledge which is an extension of $\tau$-Bench, where task success requires retrieving, reasoning over, and applying knowledge from a natural-language corpus. It include just one domain $\tau$-Banking with 698 LLM-generated documents with human supervision. Then the paper runs evaluation at 6 different models with different retrieval models or settings to showcase even the best models yet cannot achieve satisfactory performance, with some efficiency and error analysis.

**Compliance With Llm Reviewing Policy:**

Affirmed.

**Final Justification:**

I increase my score from 2 to 3 due to more clarification but the presentation can be largely refined and more experimental results required to show the value of the benchmark.

**Key Questions For Authors:**

See above

**Limitations:**

Yes

**Strengths And Weaknesses:**

Strengths:

1. the targeted problem is valuable, writing is easy-to-understand.
2. the benchmark is scalable due to minor human supervision.

Weakness

1. the contribution is limited and incremental. For example, lots of similar problems are also mentioned in $\tau$-Bench, such as policy constraints, and dynamically retrieve such constraints during interactions; discoverable tools are similar with tool retrieval / selection in tons of existing work.
2. there is just one domain in proposed benchmark, it is slightly overclaimed as $\tau$-Knowledge.
3. the writing and experiments can be better refined. ie.., a table to show direct comparsion with existing tool benchmarks or agent; benchmarks; the performance of more models and different retrevial settings (i.e., number of returned documents); task distribution analysis and so on.

---

> ### Author Rebuttal · Authors · 2026-03-28
>
> Thank you for your review. Below we address your concerns:
>
> > the contribution is limited and incremental.
>
> The contribution of τ-Knowledge lies in the integration of capabilities that in isolation are familiar, but whose combination directly models one of the most pervasive workflows. In domains like customer support and IT troubleshooting, millions of interactions occur daily in which agents must (1) search, interpret, and act on documentation over the course of (2) long-horizon, multi-turn conversations. To our knowledge, there does not exist benchmark that models this.
>
> This approach has clear precedent. SWE-Bench integrates function-level code generation (HumanEval, MBPP) and bug-fixing (Defects4J); WebArena similarly extends single-page interactions (MiniWoB++) into multi-step tasks. In both cases, these combinations expose failure modes invisible when those abilities are evaluated separately. τ-Knowledge plays the same role for **knowledge retrieval** and **conversational tool use**. Our error analysis (Section 6.2) confirms this directly: the dominant failures: search inefficiency (23%), complex cross-document interdependencies (14.5%), implicit subtask-ordering violations (5%), and overtrust in user assertions (4%), arise from the interaction between retrieval, reasoning, and conversation. We have revised the paper to better position τ-Knowledge this way.
>
> We also note that τ-Knowledge is substantially more difficult than other tau-bench domains. The best pass^1 on τ-Banking is 20%+, vs 70%+ on others. Realistic benchmarks that challenge SOTA reasoning models are difficult to procure + require significant manual expertise, which we view a major contribution of this work.
>
> > For example, lots of similar problems are also mentioned in $\tau$-Bench, such as…
>
> We want to highlight:
>
> * In the original τ-Bench + τ2-Bench domains, constraints are not dynamically retrieved. The entire policy is loaded into the agent's system prompt as a single static document at the start of the conversation. In τ-Knowledge, domain info and policies are distributed across ~700 interconnected documents.
>
> * We agree there is a conceptual connection to tool retrieval/selection, but discoverable tools introduce a key difference. Existing tool-retrieval benchmarks (e.g., ToolBench, API-Bank) evaluate an agent's ability to select from a large pre-defined tool catalog. In τ-Knowledge, discoverable tools are not visible to the agent at all until it retrieves the document that references them. The agent's action space expands dynamically as a function of retrieval behavior, making retrieval and reasoning directly coupled to task success. To our knowledge, no existing retrieval benchmark evaluates this coupling within a multi-turn conversational setting.
>
> We have updated our paper to make the conceptual differences between τ-knowledge and its predecessors clearer. Thanks for helping us sharpen this distinction!
>
> > it is slightly overclaimed as -Knowledge.
>
> Our benchmark follows the spirit of prior Tau-Benches, where domains serve as concrete instantiations of a broader problem type, and the value comes from the quality and rigor of the tasks rather than from domain count alone. We believe the depth of tau-banking (~700 interlinked documents, 51 discoverable tools, 97 tasks averaging 18.6 gold documents each) is an appropriate scope. That said, we agree that the evaluation novelty is overclaimed, and have made changes to calibrate this better!
>
> > the writing and experiments can be better refined…
>
> We appreciate this comment. We want to highlight the below details/changes to address your concerns:
>
> * Comparison with existing benchmarks: Section 2 provides detailed textual comparison across 6 agent and tool-use benchmarks and 9 retrieval and knowledge-centric benchmarks. That said, we agree that a structured comparison table would improve clarity, and have added one in the revised version.
> * More models: We evaluate 10 model–reasoning configurations across 5 providers. In the revised version, we have additionally added results for GPT-4.1 and GPT-4o (Appendix F, Table 12), further broadening coverage.
> * Different retrieval settings: We include extensive retrieval hyperparameter ablations in Appendix F. Specifically, Table 5 reports the impact of varying k ∈ {5, 10, 20} across all three retriever families. We additionally ablate reranker presence (Table 3), grep tool inclusion (Table 4), and write-tool access in the terminal configuration: over 12 retrieval variants in total.
> * Task distribution analysis: Table 1 includes domain summary statistics, three full task samples in Appendix B illustrate the diversity of task types, and we have a qualitative error analysis in Section 6.2 identifying the four most common failure categories. To the reviewer’s point, we are adding explicit failure frequency percentages to better quantify the distribution of failure modes, as well as added sections to better highlight workflow distribution.

---

> > ### Author Rebuttal · Reviewer_8hyr · 2026-04-05
> >
> > Thanks the authors for the clarification, I decide to slightly increase my score since I think this paper need to be refined significantly in terms of both presentation and experiments.

---

> > > ### Author Response · Authors · 2026-04-05
> > >
> > > We are very glad that our response led you to increase your score, thank you for making this a constructive process. We want to highlight the points brought up by your original review regarding *presentation* and *experiments* to ensure that your final decision is made with full context!
> > >
> > > ### **Experiments**
> > >
> > > **Models.** Our paper evaluates 10 model–reasoning configurations across 5 providers: GPT-5.2 (high + none), Claude-4.5-Opus (high), Claude-4.5-Sonnet (high), Gemini-3-Pro (high), Gemini-3-Flash (high), and DeepSeek-V3.2 (enabled + none). The revision adds GPT-4.1 and GPT-4o (Appendix F, Table 12), bringing the total to 12 configurations across 6 model families.
> > >
> > > **Retrieval configurations.** Each model is evaluated under 7 distinct settings:
> > > * Dense retrieval with two embedding models (text-embedding-3-large, Qwen3-embedding-8B)
> > > * Sparse retrieval (BM25)
> > > * Terminal use (freeform shell access)
> > > * Golden retriever (gold documents in-context, isolating reasoning from retrieval)
> > > * No-knowledge baseline (validating tasks genuinely require retrieval)
> > > * Long-context baseline (full ~200K-token KB in prompt, testing whether non-gold documents act as confounders)
> > >
> > > This yields over 70 unique (model × retrieval) combinations in the main experiments alone.
> > >
> > > **Retrieval hyperparameter ablations (Appendix F).** We ablate across four axes:
> > > * Reranker: with/without pointwise LLM reranker × all retrievers × grep settings (Table 3)
> > > * Grep tool: with/without regex search alongside each retriever × reranker settings (Table 4)
> > > * Retrieval set size: k ∈ {5, 10, 20} across all three retriever families (Table 5)
> > > * Write access in terminal: read-only vs. read-write, with qualitative analysis of model behavior
> > >
> > > This amounts to 12+ additional retrieval variants beyond the main configurations.
> > >
> > > **Metrics.** We report 6 complementary metrics: pass^1–4 (accuracy + reliability), document recall, action recall (continuous proxy beyond binary pass/fail), cost ($/task), duration, and median turn time.
> > >
> > > **Statistical rigor.** Three full pairwise bootstrap significance tables (Tables 7–9, 1,000 iterations) covering retrieval configs on pass^1, retrieval configs on action recall, and models on pass^1.
> > >
> > > We believe this is among the more thorough evaluations in the agentic benchmark space. We would have loved to include more ablations from the appendix in the main paper, but encountered space constraints. The reviewer's suggestions for **more models** and **different retrieval settings** are areas we had already invested significantly in; and hopefully this summary makes that coverage clearer!
> > >
> > > ### **Presentation**
> > > **Structured comparison table.** The reviewer suggested a table comparing τ-Knowledge with existing benchmarks. Section 2 provided detailed textual comparison across over 15 prominent benchmarks (SWE-Bench, WebArena, ToolBench, API-Bank, GAIA, τ-Bench, BEIR, MTEB, BRIGHT, CORAL, mTRAG, among others), but we agree a structured table improves clarity — this has been added in the revised version. Page constraints made it difficult to include additional figures in the submitted version.
> > >
> > > **Task distribution and domain statistics.** Table 1 reports detailed domain statistics: 698 documents across 21 categories, 51 discoverable tools, 97 tasks averaging 18.6 gold documents and 9.52 required tool calls each (range: 1–33). Appendix B provides three complete task samples spanning multi-constraint product recommendation (Sample 1), multi-step procedural policy execution (Sample 2), and operation sequencing with cross-action dependencies (Sample 3). Appendix C includes a full annotated conversation trace. In the revision, we have added explicit failure-mode frequency percentages to Section 6.2 (~23% search inefficiency, ~14.5% cross-document interdependencies, ~5% subtask ordering, ~4% overtrust) and additional sections highlighting workflow distribution across task types.
> > >
> > > **Additional revisions.** We have sharpened the positioning of τ-Knowledge relative to prior τ-Bench domains (Section 1), clarified the distinction between discoverable tools and standard tool retrieval/selection, and calibrated claims about evaluation novelty. We have also added the comparison table referenced above, two additional model evaluations (GPT-4.1, GPT-4o), and improved discussion of how retrieval configuration affects solution efficiency.
> > >
> > >
> > > Hopefully this context helps address your concerns regarding the presentation and experiments posited in the original review. If there are any additional concerns, we would not hesitate to address it. We believe strongly in our evaluation, and that hillclimbing it will lead to knowledge-based systems that better interact with users!

---

### Official Review · Reviewer_Hrb7 · 2026-02-26

**Soundness:** 3
**Presentation:** 3
**Significance:** 3
**Originality:** 3
**Overall Recommendation:** 4
**Confidence:** 3

**Summary:**

This paper presents τ-Knowledge, a benchmark designed to test how well language agents can handle unstructured knowledge during conversations. Building on the previous τ-Bench, the authors introduce a new "τ-Banking" domain where agents act as customer support representatives. To succeed, these agents must read through a large corpus of documents, retrieve the correct information, and interestingly, discover new tools that are only defined within those texts. The evaluation uses a simulated user and measures success based on whether the agent correctly updates a database. The authors tested several frontier models. The results show that the task is incredibly difficult. The experiments suggest that the main bottleneck isn't finding the right document, but rather the agent's ability to reason over that information once retrieved.

**Compliance With Llm Reviewing Policy:**

Affirmed.

**Key Questions For Authors:**

1. Is it possible to run a small human evaluation on a subset of these tasks (e.g., 20 tasks)? Knowing the human ceiling can help readers better understand whether the low model performance is due to reasoning failures or task ambiguity.

2. Why is the "Action Recall" metric relegated to the appendix?

**Limitations:**

yes

**Strengths And Weaknesses:**

Strengths

- This work presents a novel "discoverable tools" mechanism: The idea that agents must discover their own tools by reading documentation is a very compelling design choice.

- By feeding the agent the perfect document and seeing that performance still caps at around 35%, the authors provide proof that the problem lies in reasoning and knowledge utilization, not just in the retrieval step. This offers a valuable insight to the community.

Weaknesses

- The benchmark currently only covers a single domain (banking). While the depth is good, it is a step back from the multi-domain approach of the original τ-Bench. It is difficult to say if the findings, such as the dominance of the agent model over the retrieval method, will generalize to other fields like legal or medical support.

- The primary novelty is adding a knowledge base and discoverable tools to an existing framework. While this is a meaningful extension, the contribution is closer to a new domain/dataset for an existing benchmark than a new evaluation paradigm. The paper would benefit from a more explicit discussion of what is architecturally new versus inherited.

---

> ### Author Rebuttal · Authors · 2026-03-28
>
> Thanks for your thoughtful review. We are glad that you found our discoverable tools mechanism novel, and the importance of our golden documents setup to causally isolate performance factors. Below we address your concerns:
>
> > While the depth is good, it is a step back from the multi-domain approach of the original τ-Bench. It is difficult to say if the findings will generalize…
>
> In Section 6.2, we show that the sources of difficulty in τ-Knowledge are not specific to the banking domain. The dominant failure modes, search inefficiency + unwarranted assumptions (23%), complex documents interdependency (14.5%), failure to respect implicit subtask ordering (5%), and overtrusting user assertions (4%), are all domain-agnostic reasoning failures. Agents fail not because banking concepts are uniquely confusing, but because they struggle to coordinate retrieval with multi-hop reasoning across documents, to infer ordering constraints between actions, and to verify claims against system state before acting.
>
> But we fully agree that expanding to multiple domains is valuable. Our construction framework is designed to be extensible and is fully open-source: τ-Knowledge provides the primitives and tooling needed for the τ-Bench community to build new knowledge-intensive conversational domains.
>
> > While this is a meaningful extension, the contribution is closer to a new domain/dataset for an existing benchmark than a new evaluation paradigm. The paper would benefit from a more explicit discussion of what is architecturally new versus inherited.
>
> We agree with the reviewer, and appreciate this feedback. The framing of τ-Knowledge as a new evaluation paradigm was stronger than intended.
>
> τ-Knowledge combines two axes of evaluation that have been studied extensively in isolation but rarely evaluated together: conversational, multi-turn human-agent interaction and knowledge retrieval over unstructured corpora. This is a combination that is the norm in real-world deployments like customer support and technical troubleshooting, but to our knowledge is not captured by existing benchmarks.
>
> We have made this framing clearer in the revised paper by: (1) Softening language throughout to position our contributions as combining two well-studied axes of evaluation; (2) adding an explicit paragraph in the introduction that positions τ-Knowledge as retrieval-mechanism-agnostic, enabling comparison across existing search strategies; (3) more clearly disambiguating τ-Banking (the domain and benchmark) from τ-Knowledge (the evaluation setting); (4) expanding the benchmark construction section with a dedicated review stage (Stage 5), highlighting the human verification effort as a core contribution.
>
> We also note one additional contribution beyond discoverable tools and the knowledge base: a flow-based user simulation that specifies conditional exact utterances at evaluation-critical junctures, making the simulator more reliable and helping produce the low critical-error rate reported in Section 6.1.
>
> > Why is the "Action Recall" metric relegated to the appendix?
>
> Action Recall is relegated to the appendix because it is an imperfect measure of partial progress that we did not want to overemphasize. While it captures whether the agent performed the required actions, it can be misleading: an agent may complete a large fraction of required actions yet still fail the task entirely (e.g., performing steps in the wrong order, or missing a single critical action that invalidates downstream steps), and an agent with 100% Action Recall can still fail by performing additional actions that should not have been taken.
>
> > Is it possible to run a small human evaluation on a subset of these tasks (e.g., 20 tasks)?
>
> We agree that a small human evaluation would be interesting! However, we do not think it is necessary for this paper’s core contribution, which is to introduce and validate a benchmark for challenging knowledge-intensive support tasks. Many of these tasks are solvable by experienced human agents, but doing so often relies on substantial domain training and familiarity with the knowledge base. Our goal here is therefore not to compare models against untrained humans or to establish human-level performance, but to evaluate model performance on difficult cases that arise in realistic support settings. To address the more immediate concern of task validity and ambiguity, we instead conducted a formal task review to verify that each task has a valid path to success.
>
> Once again, thanks for your thoughtful review, and we look forward to engaging in further discussion to resolve any further concerns. We believe strongly in our evaluation, and that hillclimbing it will lead to knowledge-based systems that better interact with users!

---

> > ### Author Rebuttal · Reviewer_Hrb7 · 2026-04-06
> >
> > The authors have adequately addressed my concerns. The clarification on domain-agnostic failure modes and the revised framing of the contribution are convincing. I maintain my original score.

---

### Official Review · Reviewer_nWJL · 2026-03-13

**Soundness:** 3
**Presentation:** 3
**Significance:** 4
**Originality:** 3
**Overall Recommendation:** 5
**Confidence:** 5

**Summary:**

The paper suggests an extension to the Tau-Bench framework that tests whether agents can incorporate NL knowledge and act accordingly.

During the last year, we have seen many attempts to integrate agents into real production-level businesses; however, it is very difficult to earn the trust of the users and the agent builder. I am a true believer in knowledge and policy for agents, and think that this is the main requirement for agents to succeed in enterprise adoption.

I am quite familiar with Tau-Bench and other agent benchmarks, so I will not write a long review. I really like the paper, and I think it is a major contribution to the community. I do think that if the benchmark can fit both a computer-use agent and an API-based agent, with less focus on the conversation part, it can have a major impact.

**Compliance With Llm Reviewing Policy:**

Affirmed.

**Final Justification:**

The author addresses my questions. As an expert in the domain, I believe the work is worthy of publication.

**Key Questions For Authors:**

1. How did you let the agent know the policy and knowledge? Did you just put it in the system prompt? Rag?
2. Can your approach be extended to computer-using agents, or even something that is less conversational?
3. Please test, even on a small subset of samples, a true enterprise-ready multi-agent system with policy.

* I know that 3 might be difficult, but it is super important for me to be convinced that it is not just a problem with the vanilla agent you tested

**Limitations:**

Yes, the authors discussed the limitations in detail

**Strengths And Weaknesses:**

Strengths
1. The topic is very important to the ML community, and the paper addresses one of the main problems that agents are struggling with: the knowledge.
2. I would feel comfortable running an agent in my organization that got super high accuracy on these benchmarks, as I know it will follow my rules, instructions, and policies.
3. I am actively building agents, and I believe this benchmark has the potential to take the adoption of agents in real settings one step ahead


Weaknesses
1. The paper presentation is fine, but can definitely be improved. As the main contribution of the paper is the benchmark itself, I would put more focus on what kind of things the benchmark is testing and how exactly it tests them.
2. You did not test a real enterprise-ready multi-agent system that knows how to work with policy, knowledge, etc. This is a major concern as the vanilla agents were not designed to tackle this kind of problem.
3. The paper is missing some important references and a short discussion on the sister domains
For example: Web agent and computer using agents (e.g., AgentE, Claude computer use, IBM CUGA, Open AI Operator),  and benchmarks that test policy (e.g., SafeCUA, ST-WebBench) and knowledge.
The domain of computer-using agents is closely related to yours, so you should address it properly (no experiments, just cite and explain the difference).

---

> ### Author Rebuttal · Authors · 2026-03-28
>
> Thank you for your thoughtful review. Given your familiarity with τ-Bench and hands-on experience building agents in production, we are especially glad that you view τ-Knowledge as a major contribution to the community, and that high accuracy on this benchmark would give you confidence in deploying an agent in your organization. Below we address your remaining concerns.
>
> > As the main contribution of the paper is the benchmark itself, I would put more focus on what kind of things the benchmark is testing and how exactly it tests them.
>
> This is a good point, and we have made targeted revisions to make this more clear. Specifically, we: (1) Added failure mode frequency distributions to the qualitative analysis (Section 6.2), quantifying how often each category of error occurs across all tested configurations, giving readers a concrete picture of what the benchmark tests. (2) Added a dedicated "Disentangling Search from Information Usage" analysis (Section 5) that explicitly separates retrieval failures from reasoning failures. (3) Expanded the benchmark construction section with a new Review stage (Stage 5) describing the independent auditing protocol, including manual trajectory verification and post-experiment re-auditing. (4) Restructured the main results table to show Gold-relative performance deltas, making the retrieval-vs-reasoning gap immediately visible. (5) Simplified the formal Dec-POMDP presentation in favor of a more accessible description of the agent's task.
>
> > You did not test a real enterprise-ready multi-agent system that knows how to work with policy, knowledge, etc.
>
> We appreciate the reviewer's concern and address it on three fronts:
>
> * We do evaluate a basic form of multi-agent pipeline: our retrieval configurations include a pointwise LLM reranker stage (Appendix D.1), which introduces a secondary LLM call (with sota reranker LLM) to re-score and filter retrieved documents before they reach the primary agent. This represents a basic but meaningful form of multi-stage, multi-model orchestration known to be used in production systems. However, we found that adding the reranker did not improve performance (and in some cases slightly degraded it), suggesting that the bottleneck is not in retrieval quality alone.
>
> * In terms of agentic search capabilities at least, our analysis shows that an optimal search agent would not saturate this benchmark. Even with the exact set of task-critical documents directly in context, the best model (Claude-4.5-Opus, high reasoning) achieves only 39.69% pass^1. This demonstrates that a substantial portion of the difficulty in τ-Knowledge lies in the underlying model's ability to reason over complex, interdependent policies and apply them correctly through tool calls. No retrieval or orchestration architecture can outperform an oracle retriever, so multi-agent designs would face the same reasoning bottleneck.
>
> * Overall though, we agree that evaluating more sophisticated agent architectures is a very valuable direction. τ-Knowledge is designed to be agent-architecture agnostic: any system (multi-agent, single-agent, or rule-based) can be evaluated through the same tool-use interface. We openly encourage teams that have developed enterprise-grade multi-agent systems to utilize our benchmark: our code is fully open source, and we maintain an online leaderboard to help people share their results. We have added a paragraph to the Limitations section noting that our evaluation focuses on single-model agent architectures and that evaluating production-grade orchestration systems is a promising direction for future work.
>
> > The paper is missing some important references and a short discussion on the sister domains…
>
> Thank you for pointing this out. We have expanded our Related Work section to discuss the broader landscape of agent evaluation, including the web and computer-use agents work that you have pointed out (e.g., AgentE, Claude Computer Use, IBM CUGA, OpenAI Operator) and benchmarks that evaluate policy adherence in these settings (e.g., SafeCUA, ST-WebBench).
>
> > How did you let the agent know the policy and knowledge?
>
> The agent's system prompt contains only general behavioral guidelines, such as identity verification procedures, escalation rules, and instructions for how to access the knowledge base (Appendix D). All procedural policies, product details, and discoverable tool signatures resides in the external knowledge base of ~700 documents. The agent accesses this corpus at inference time through sets of retrieval tools, including dense semantic search (text-embedding-3-large, Qwen3-embedding), sparse lexical retrieval (BM25), and freeform terminal-based exploration using standard Unix utilities (grep, cat, find). Full system prompts are provided in Appendices D.1 and D.2.
>
> We look forward to engaging more with you to clear up any additional questions/concerns about our work, and are happy to answer any more questions as we believe strongly in our work!

---

> > ### Author Rebuttal · Reviewer_nWJL · 2026-04-03
> >
> > The author addresses my questions.
> > As an expert in the domain, I believe the work is worthy of publication.
> > I ask the authors to add the appropriate references and to revise the manuscript flow so it will be better understood, and that the contribution will be accessible to all readers.

---

> > > ### Author Response · Authors · 2026-04-05
> > >
> > > We are glad to have resolved all your lingering concerns, and your expert validation of our work is greatly appreciated and valued. Thank you for making the review process a constructive experience. The final version of the paper will contain the appropriate references and framing adjustments to make the contribution more accessible.

---

### Decision · Program_Chairs · 2026-04-30

**Decision:**

Accept (regular)

**Comment:**

This paper addresses an important problem related to agent faithfulness: whether an agent can faithfully incorporate natural language knowledge and act accordingly.

The main concerns relate to presentation, the single-domain setting, and its positioning relative to τ-Bench. Two out of three reviewers are fully convinced by the authors’ rebuttal. I agree that the presentation and the single-domain scope are limitations. However, given the importance of the problem and the reasonable (though not extensive) evaluation framework, I believe this work can be useful to the community, particularly for researchers and practitioners working in enterprise settings.

I expect the authors to improve the clarity of the presentation and extend the evaluation, as suggested by the reviewers.